# High-Performance Photodetectors Based on Nanostructured Perovskites

**DOI:** 10.3390/nano11041038

**Published:** 2021-04-19

**Authors:** Chunlong Li, Jie Li, Zhengping Li, Huayong Zhang, Yangyang Dang, Fangong Kong

**Affiliations:** 1Key Laboratory of Pulp and Paper Science & Technology of Ministry of Education, Qilu University of Technology, Shandong Academy of Sciences, Jinan 250353, China; knje@163.com (Z.L.); headingzhy@126.com (H.Z.); 2International College of Optoelectronic Engineering, Qilu University of Technology, Shandong Academy of Sciences, Jinan 250014, China; jiejie19831228@163.com; 3Shandong Provincial Key Laboratory of Laser Polarization and Information Technology, School of Physics and Physical Engineering, Qufu Normal University, Qufu 273100, China; 4Tianjin Key Laboratory of Molecular Optoelectronic Sciences, Department of Chemistry, School of Sciences, Collaborative Innovation Center of Chemical Science and Engineering, Tianjin University, Tianjin 300072, China

**Keywords:** nanostructured perovskites, high-performance photodetector, different dimensions

## Abstract

In recent years, high-performance photodetectors have attracted wide attention because of their important applications including imaging, spectroscopy, fiber-optic communications, remote control, chemical/biological sensing and so on. Nanostructured perovskites are extremely suitable for detective applications with their long carrier lifetime, high carrier mobility, facile synthesis, and beneficial to device miniaturization. Because the structure of the device and the dimension of nanostructured perovskite have a profound impact on the performance of photodetector, we divide nanostructured perovskite into 2D, 1D, and 0D, and review their applications in photodetector (including photoconductor, phototransistor, and photodiode), respectively. The devices exhibit high performance with high photoresponsivity, large external quantum efficiency (EQE), large gain, high detectivity, and fast response time. The intriguing properties suggest that nanostructured perovskites have a great potential in photodetection.

## 1. Introduction

Photodetectors—the vital components of modern imaging and communication systems—have been playing an increasingly important role in modern industrial production, basic scientific research, space development, ocean exploration, military and national defense, environmental protection, medical diagnosis, transportation, and other fields. For example, ultraviolet photodetectors can be used in ozone sensing, flame sensing, etc. [1,2,3]. The visible photodetectors can be used in biological sensing, video imaging, and convert communications [4,5,6,7]. Infrared photodetectors can be used as infrared night vision. [8,9,10] The THz photodetectors can be used in the security detection of customs, airports, and other special occasions [11,12,13]. Therefore, the further research of photodetectors is of great significance. A semiconductor, which is essential for a photodetector, can absorb the incident photons and generate electron and hole pairs. In the presence of a built-in or applied electric field, electric current is generated when the electrodes extracted and collected photogenerated carriers. In order to obtain a high-performance photodetector with high sensitivity and fast response, the semiconductor needs to have effective charge collection, low trap state density, and high carrier mobility. Till now, a large variety of semiconductor materials have been used for constructing photodetectors, including Si [14,15,16], carbon nanotubes [17,18], group II–VI and III–V compounds [19,20,21], and remarkable progress has been made in improving the detection performance. Epitaxial growth technology with stringent conditions, which is the most commonly used for synthesis, hampers their commercial application [22]. Therefore, it is of great significance to exploit candidates to reduce production cost and improve photodetector performance.

Recently, perovskites with a typical formula of ABX_3_ has attracted wide research interest in the photodetector field [23,24,25]. For ABX_3_, A is a monovalent cation (CH_3_NH_3_^+^ (MA), Cs^+^, etc.), B is the divalent metal cation (Pb^2+^, Sn^2+^, etc.), and X is a halide ion (Br^−^, Cl^−^ and I^−^). Many properties of perovskite make it an ideal material for photodetectors. For example, ambipolar transport, low density of defects and traps within bandgap can effectively reduce the charge recombination, and thus improve the performance of photodetectors. [26] The absorption spectra can cover the UV–Visible–Infrared region by facile halide substitution, which is desirable for broadband photodetector applications [27,28]. High absorption coefficient and direct bandgap result in a fast photoresponse in a very thin perovskite layer because of short transmission distance of photongenerated carriers [29,30]. In addition, low cost and easy preparation remove obstacles in future mass preparation. For example, MAPbI_3_ film-based photodetectors with a metal–semiconductor–metal (MSM) structure exhibited a broad photoresponse range from 310 nm to 780 nm, with a photoresponsivity of 3.49 A W^−1^ and external quantum efficiency (EQE) of 1.19 × 10^3^% [31]. However, there is an inherent paradox to simultaneously possess both low dark currents and high photocurrents. The former requires a large number of defects or barriers that appear in polycrystalline film to inhibit the transmission of thermally excited carriers [32,33,34], while the latter requires single crystals with good crystallinity for effective charge transfer [35,36,37]. Compared with photodetectors based on polycrystalline film and bulk crystals, nanostructured photodetectors exhibit superior performances. The large surface-to-volume ratios of nanostructures result in longer photocarrier lifetime, which is conducive to higher sensitivity and responsivity. In addition, the reduced dimension shortens the carrier transmission time and improves the response speed [19,38,39,40,41]. Therefore, the performance of photodetector based on nanostructured perovskites will be better. For example, a high responsivity of 1294 A W^−1^ with a ultrahigh detectivity of 2.6 × 10^14^ Jones was obtained in α-CsPbI_3_ nanowire-based photodetector [42]. In addition, an ultrahigh response speed (19/25 µs) was obtained in a photodetector based on atom-thin 2D CsPbBr_3_ nanosheets [43]. An ultrahigh EQE over 10^7^% was demonstrated by a phototransistor based on CsPbI_3-x_Br_x_ quantum dots (QDs)/monolayer MoS_2_ heterostructure [44]. All these are enough to show that the photodetectors based on nanostructured perovskites have more advantages in ultrahigh responsivity and ultrafast response speed. There have been some reviews on nanostructured perovskite-based photodetectors. Gu et al. [45] focus on the effect of elemental composition and dimensionality of the perovskite materials on photodetector performance. Wang et al. [46] systematically summarized the synthesis, optoelectronic properties, and performance of photodetectors based on low-dimensional perovskites. Here, more emphasis is placed on the effect of the device structure and the dimension of nanostructured perovskites on the device performance.

The key parameters of photodetectors are shown in Table 1.

In this study, we will review research results in nanostructured perovskite-based photodetectors, focusing on the photodetection performance and potential mechanism. It is well known that the electrons of nanostructured perovskite are quantum confined in three directions for 0D QDs, two directions for 1D nanowires (NWs), and one direction for 2D nanosheets [45]. The band structures of the nanostructured perovskite could be heavily influenced by the quantum size effect induced by dimensionality constraint, and thus deeply affect the optoelectronic properties. Therefore, we discussed the performance of perovskite photodetectors based on 1D, 2D, and other nanostructured perovskites, respectively. Finally, a brief summary and outlook will be given to enhance the performance of the perovskite-based photodetectors.

## 2. Photodetectors

Perovskite-based photodetector devices can be divided into two categories, photovoltaic and photoconductive photodetectors. According to the spatial layout of the photoactive medium and electrodes, perovskite-based photodetector devices can be further divided into vertical type and lateral type. In general, vertical photodetectors provide fast response and low driving voltage because of the small electrode spacing with a short carrier transit length; in contrast, lateral photodetectors show slow response and high driving voltage due to their large electrode spacing. For photovoltaic photodetector, or photodiode (Figure 1a), the device structure is similar to that of solar cell configuration. Photodiodes based on perovskite polycrystalline films or single crystals are widely reported, but those based on nanostructured perovskites are rarely reported. Photodiodes typically rely on PN junction, which can provide a built-in electrical field at the junction interface to aid the electrons and holes to transport in opposite directions toward electrodes. Owing to the junction barrier at the interface, photodiodes exhibit low dark current and large detectivity. However, they suffer from low responsivity and external quantum efficiency (EQE ≤ 100%). As for photoconductive photodetector, it can be further divided into photoconductor (Figure 1b) and phototransistor (Figure 1c). Compared with photovoltaic photodetector, photoconductive photodetector exhibits high responsivity, EQE (beyond 100%) and large gain. External voltage leading to multiple electrical carriers recycling per single incident photon should be responsible for the large gain [47,48,49,50]. However, large gain, in turn, usually results in a slow response speed because both the response time and the gain are determined by the carrier lifetime. Therefore, the intrinsic contradictions between the responsivity and response speed always exist. One solution is to fabricate phototransistor by adding gate electrode (Si) and dielectric layer (SiO_2_) to the photoconductor. The charge transport can be controlled by applying a gate voltage. It is demonstrated that phototransistor can simultaneously enhance the photoresponsivity and exhibit an ultrafast photoresponse speed [51].

### 2.1. Photoconductor

#### 2.1.1. Photoconductor Based on 2D Perovskites

Among the various perovskite compositions, CsPbX_3_ and CH_3_NH_3_PbX_3_ (MAPbX_3_) (X = Br, I) have attracted more attention in photodetection. Song et.al report, [43] for the first time, the preparation of atom-thin 2D CsPbBr_3_ nanosheets and their high-performance in flexible photodetector with solution treatment. The UV–vis absorption spectrum (Figure 2a) of the CsPbBr_3_ nanosheets exhibited a favorable absorption capability and a direct bandgap of about 2.32 eV. The schematic of a flexible photodetector device based on the as-fabricated CsPbBr_3_ nanosheet is shown in Figure 2b. The flexible photodetector exhibited a high on/off ratio (>10^3^, Figure 2c), which indicated a good light-switching behavior, high responsivity of 0.25 A W^−1^(Figure 2d) and peak EQE value of 53% (Figure 2e). The high switching ratio as shown in Figure 2f results from large absorption coefficient of the perovskites. The rise and decay times were 19 and 25 µs (Figure 2g), respectively, which are much shorter than the previously reported [52]. The high response speed can be attributed to the high carrier transport speed caused by high crystal quality and atomic 2D plane of the CsPbBr_3_ nanosheets. A fluctuation that was <3% after bending for 10,000 times indicates high flexibility (Figure 2h). In addition, a fluctuation of less than 2.6% after 12 h of exposure indicates excellent stability (Figure 2i). These results show that there is a great potential for CsPbBr_3_ nanosheets in high-sensitivity detectors.

Qin et al. [53] successfully prepared high-quality MAPbI_3_ with the morphologies of nanowires (NWs) and nanoplates by a simple solution immersing method. Schematic diagrams of photodetectors based on nanowire and nanoplate are shown in Figure 3a,b. The stability was tested by dozens of cycles under different illumination (Figure 3c,d). Both the photocurrent and switching ratio were maintained well, and the switching ratio was well controlled by adjusting the applied illumination intensity. The on/off ratio of the nanowire-based MAPbI_3_ photodetector reached 314, while the on/off ratio of nanoplate-based photodetector up to 1210, which was several orders of magnitude higher than that of polycrystalline film photodetector [54]. The nanoplate-based devices usually exhibit a relatively better performance than nanowire-based devices due to the higher crystal quality with smoother surface and more regular shapes. It suggested that higher performance could be expected by further improving the crystal quality.

Due to low responsivity induced by poor charge transport of perovskites, many studies have combined perovskites with high mobility materials to increase responsivity. For example, Li et.al [55] constructed a photodetector based on CsPbBr_3_ nanosheet/carbon nanotubes (CNTs) heterojunction to improve performance. CNTs act as a transport layer with high carrier mobility, and CsPbBr_3_ nanosheets act as a photo absorber with strong absorption. The schematic diagram of the photodetector was illustrated in Figure 4a. The experiment proved that CsPbBr_3_ nanosheet/CNT (6%) composite-based photoconductor exhibits almost the best responsivity. The highest external quantum efficiency (EQE) reaches 7488% (Figure 4b) and the highest responsivity reaches 31.1 A W^−1^ (Figure 4c). The on/off ratio reaches 823 (Figure 4d). A nonlinear I–V curve demonstrates Schottky contact between CsPbBr_3_ nanosheet/CNT and electrodes. Figure 4e shows a broad LDR of the device. The I–t curves exhibit excellent reproducibility and stability (Figure 4f). The rise and decay times were 16 µs and 0.38 ms, respectively (Figure 4g). The high response speed indicates the rapid separation and efficient extraction of photogenerated carriers, which can be attributed to the improved electrical conductivity of CNTs.

#### 2.1.2. Photoconductor Based on 1D Perovskites

Deng et al. [56] first reported photodetectors based on single-crystalline MAPbI_3_ microwire (MW) arrays (Figure 5a). The device demonstrated an obvious response to visible light, but was rather insensitive to the UV and NIR light. (Figure 5b,c). High responsivity of 13.57 A W^−1^ (Figure 5d), high detectivity of 5.25 × 10^12^ Jones and broad LDR (Figure 5e) were achieved in the MW array-based photodetectors. Compared with thin film-based photodetector, the MW array-based ones exhibit better stability by recording the dark and photocurrents of the photodetectors for 50 d (Figure 5f). The outstanding device performance can be attributed to the high optical absorption coefficient of MAPbI_3_ and high crystallinity of the MWs.

Li et al. [57] synthesized high crystalline MAPbI_3_ MWs arrays by a solution-based blade coating and solvent recrystallization method. The corresponding schematic diagram of the as-fabricated photodetector was illustrated in Figure 6a. Apparently, the device exhibits a significant response to the UV and visible light, but is insensitive to the near-IR. (Figure 6b). The energy band diagram was shown in the inset of Figure 6c. Photoresponsivity of 0.04 AW^−1^, on/off ratio of 0.84 × 10^4^ and specific detectivity of 0.6 × 10^12^ Jones are shown in Figure 6e. The rise and decay time is 178/173 µs (Figure 6f). It is worth noting that the performance of the as-fabricated device has been significantly improved. The main reasons may be as follows: (i) High crystallinity and suitable surface roughness of the MAPbI_3_ MWs are beneficial for the transport of carriers. (ii) The MAPbI_3_ MWs and Ag electrodes form a stable and available Ohmic Contact. The results demonstrate that the MAPbI_3_ MWs have a good application prospect in high-performance photodetectors.

By reacting Pb-containing precursor NWs with MABr and HBr in an organic solvent, Zhuo et al. [58] successfully prepared porous MAPbBr_3_ NWs. The UV/Vis absorption spectrum (Figure 7a) indicates a favorable absorption capability with a direct band gap about 2.22 eV. A good ohmic contact was demonstrated by the typical linear and symmetrical I–V curve (Figure 7b). The on/off ratios of 61.9 suggested a good response to the light intensity. As shown in Figure 7d, the rise and decay times were 0.12 s and 0.086 s, respectively. The excellent photoelectric properties are mainly attributed to their unique 1D porous geometry, numerous active sites, and outstanding light absorption.

Deng et al. [59] reported high-quality single-crystalline MAPb(I_1−x_Br_x_)_3_ (x = 0, 0.1, 0.2, 0.3, 0.4) NWs with an absorption spectrum ranging from 680 to 780 nm by modifying I/Br ratio. The schematic diagram of the NWs-based device is shown in Figure 8a. The NWs-based photodetectors demonstrated an ultrahigh responsivity of 1.25 × 10^4^ A W^−1^ due to the high-quality crystal structure of NWs (Figure 8c). In addition, other excellent figure-of-merit parameters were also obtained by the device, such as 3 dB bandwidth (0.8 MHz), large detectivity (1.73 × 10^11^ Jones), LDR of 150 dB (Figure 8d), maximum G of 36,800 and robust stability. The high performance could be attributed to the long carrier lifetime and high carrier mobility in high-crystalline MAPbI_3_ NWs. Thus, a variety of high-performance integrated optoelectronic devices could be fabricated by the NW arrays.

Tang et al. [60] successfully synthesized CsPb(Br/I)_3_ nanorods with a facile hot-injection method. The PL and UV–vis absorption spectra indicate that the bandgap is ≈ 1.98 eV (Figure 9c). The schematic of the photodetector was illustrated in Figure 9a. The photosensitivity of the photodetector reaches 10^3^, and the rise and decay times were 0.68 s and 0.66 s, respectively (Figure 9d). Figure 9e indicated the photodetector meets well with the ohmic characteristics with a linear I–V curve. I–t curve demonstrated a remarkable high on/off ratio of 2000. The good performance of the photodetector can be related to the long lifetime and short transit time of the photocarriers in CsPb(Br/I)_3_ nanorods with large surface-to-volume ratio and high density of deep level surface trap states.

It is a common method to enhance the stability of perovskite by ligands to improve the performance of photodetector. Gao et al. [61] successfully synthesized MAPbI_3_ NWs array by optimizing one-step self-assembly method. It turned out that the devices prepared by OA (Oleic acid)-passivated MAPbI_3_ NWs have the best performance. The photodetector structure is schematically shown in Figure 10a. A broadband photoresponse range from 400 to 750 nm. The calculated responsivity is 0.45 AW^−1^ (Figure 10b). The rising and decay time are within 0.1 ms (Figure 10c). On/off ratio of 4000 reflects an excellent photosensitivity of the device (Figure 10d). Additionally, the ultralow dark currents result from low carrier density and low thermal emission (recombination) rates enable the device to detect very weak optical signals. In addition, the detectivity of 2 × 10^13^ Jones was achieved (Figure 10d). The improved performance can be attributed to the increasing of carrier lifetime after OA passivation, which can reduces the non-radiating composite centers on the surface of the NWs and gives the device more time to collect and transfer the photogenerated carriers.

To verify the influence of perovskite morphology on photodetector performance, Liu et al. [62] prepared MAPbI_3_ with various morphology including NWs, microwires, a network, and islands by inkjet printing method with proper solvent and controlling the crystal growth rate. Photoconductor based on these different crystals were fabricated and among which, the MW-based photodetector exhibited better performance, such as a switching ratio of 16,000%, responsivity of 1.2 A/W, and normalized detectivity of 2.39 × 10^12^ Jones. The reason might be a more balance between the uniformity and low defects in MW MAPbI_3_. Both the rise time and decay time are within 10 ms (Figure 11c), which indicated the ability of MW MAPbI_3_-based photodetector to detect the rapidly changing optical signal.

Chen et al. [42] synthesized α-CsPbI_3_ perovskite nanowire arrays with preferential (100) crystallographic orientation to further enhance the photodetector performance. High photoluminescence (PL) intensity (Figure 12a) and long PL lifetime (Figure 12b) demonstrated a low trap density in α-CsPbI_3_ NWs, which originates from their suppressed grain boundaries and surface defects. High-performance photodetectors based on the as-fabricated NWs were constructed. The schematic diagram of photodetector was shown in the insert of Figure 12c. The photodetector exhibits high responsivity of 1294 A W^−1^ and detectivity of 2.6 × 10^14^ Jones (Figure 12d). The rise time is 0.85 ms, and the decay time is 0.78 ms (Figure 12f). The performance can maintain 90% after 30 days demonstrated an excellent long-term stability. The high performance mainly benefits from fewer grain boundaries and ordered crystallographic orientation.

In order to reduce the toxicity of lead to future applications, Han et al. [63] fabricated lead-free all-inorganic CsSnX_3_ (X = Cl, Br, I) perovskite NW arrays on a mica substrate with the growth direction of [100] by Chemical vapor deposition. Uniform and strong PL peak suggested a high crystallinity. Furthermore, the narrow band gap of CsSnI_3_ NW array extends the optoelectronic applications of perovskites from visible to near-infrared region and the as-fabricated photodetector based on CsSnI_3_ NW array is the first reported near-infrared detector. The performance of the CsSnI_3_ NW array-based photodetector is shown in Figure 13. The maximum responsivity occurred at 940 nm, so the photodetector is irradiated with 940 nm laser (Figure 13a). The photocurrents increase with the increase of the incident intensity and excellent stability and reproducibility were shown in Figure 13b. The rise and decay time were 83.8 and 243.4 ms, respectively. The fast photoresponse can be attributed to the high-quality of CsSnI_3_ NW array with less surface states and trap centers. The responsivity and detectivity were 54 mA W^−1^ and 3.85 × 10^5^ Jones, respectively.

Li et al. [64] constructed a polarization-sensitive UV photodetector based on another all-inorganic perovskite CsCu_2_I_3_ NW (Figure 14a). Anisotropy ratio of PL intensity can be up to 3.16 (Figure 14c). The schematic diagram of a photodetector based on CsCu_2_I_3_ NW was shown in Figure 14d. As shown in Figure 14e, the device can respond to light from 230 to 350 nm. The asymmetrical *I*–*V* curves demonstrated the formation of a Schottky barrier (Figure 14f). The performance of photodetector based on CsCu_2_I_3_ NW is impressive, such as a high on/off ratio of 2.6 × 10^3^ (Figure 14g), a photoresponsivity of ~32.3 AW^−1^, a high specific detectivity of 1.89 × 10^12^ Jones (Figure 14h), and a fast response speed of 6.94/214 µs (Figure 14i). In addition, a good flexibility and stability had been demonstrated by 1000 bending cycles without no photoresponse degradation (Figure 14j).

#### 2.1.3. Photoconductor Based on Other Nanostructured Perovskites

Dong et al. [65], for the first time, fabricated a photodetector based on the all-inorganic perovskite CsPbBr_3_ nanocrystals (NCs) with synergetic effect of preferred-orientation and plasmonic effect. The schematic diagram of the device is shown in Figure 15a. Figure 15b indicated a broadband photodetection range from 300 to 520 nm. The peak responsivity value is 20.92 mA W^−1^. The increase below 520 nm is attributed to the increased concentration of electron–hole pairs. The light on/off ratio is >1.6 × 10^5^, as shown in Figure 15c. Figure 15d exhibited stable and reproducible photoresponse. The rise and decay time were 0.2 and 1.3 ms, respectively (Figure 15e).

Ramasamy et al. [52] chose red emitting CsPbI_3_ NCs to fabricate photoelectronic devices because a relatively longer radiative lifetime than the green and blue emitting in CsPbX_3_ (X = Cl, Br) NCs helps to generate large photocurrent. The schematic diagram of the CsPbI_3_-based photodetector is shown in Figure 16a. The as-fabricated photodetector exhibits a high performance, including high photosensitivity of 10^5^ (Figure 16b), reproducible response to ON/OFF cycles (Figure 16c), fast response time of 24/29 ms (Figure 16d), which make a promising application in photoelectric devices.

Algadi et al. [66] constructed nitrogen doped graphene quantum dots (GQDs)/CsPbBr_3_ NCs photodetectors, in which the GQDs act as an electron transfer layer, while the CsPbBr_3_ as light absorber. Figure 17a presents the schematic diagram of the hybrid photodetector. As shown in Figure 17b, the PL lifetime of the GQDs-passivated CsPbBr_3_ NCs is obviously decreased, demonstrating a great deal of charge transfer at the GQDs/CsPbBr_3_ interface. The heterostructure-based photodetector exhibits a higher performance than pure one, such as higher on/off ratio of 7.2 × 10^4^ (Figure 17c), higher photoresponsivity of 0.24 AW^−1^ (Figure 17d), larger specific detectivity of 2.5 × 10^12^ Jones, EQE of 57% (Figure 17e), shorter decay time as 1.16 ms (Figure 17f).

### 2.2. Phototransistor

#### 2.2.1. Phototransistor Based on 2D Perovskites

The above experimental results show that it is difficult to obtain high responsivity and fast response speed simultaneously by using photoconductive device. The phototransistor can achieve a balance between the responsivity and response speed by modulating the gate voltage.

Liu et al. [67] constructed a field-effect transistor (FET) based on 2D MAPbX_3_ nanosheet prepared by a combined solution process and vapor-phase conversion method. The schematic diagram of the FET device is shown in Figure 18a. Figure 18b shows the picture and PL mapping images of the FET. The photocurrent under dim light promises great potential as an effective photodetector. The linear I–V curves (Figure 18c) indicate the contact between 2D MAPbX_3_ nanosheet and gold electrodes is ohmic. The on/off ratio of the FET can reach up to 10^2^ (Figure 18c, inset), which can be related to strong light–matter interaction and absorptive capacity of 2D MAPbX_3_ nanosheet. I–t curve further demonstrated an effective optical switching, as shown in Figure 18d. The responsivity of the FET was calculated to be 22 AW^−1^ (Figure 18e). Rise and decay times of the FET are within 20 and 40 ms, respectively (Figure 18f). The results demonstrate that the 2D perovskite photodetector has excellent photoresponsivity and relatively fast response speed.

Lv et.al [68] fabricated a FET based on 2D all-inorganic CsPbBr_3_ nanosheet to investigate photoelectronic performance. The schematic was illustrated in Figure 19a. I–V curve with a significantly increased photocurrent (Figure 19b) demonstrates that 2D CsPbBr_3_ nanosheet has a great potential to be an efficient photodetector. The fast and reproducible on/off cycle exhibit an excellent photo switching and stability of the as-fabricated photodetector (Figure 19c). The rise time was 17.8 ms, and the decay times were determined to be 14.7 and 15.2 ms (Figure 19d).

#### 2.2.2. Phototransistor Based on 1D Perovskites

In 2014, Horvath et al. [69] reported the first MAPbI_3_ NWs-based photodetectors. Figure 20a,b showed the schematic diagram and the microscope image of the device, respectively. The linear output characteristics indicate that the contacts between MAPbI_3_ NWs and electrodes are ohmic (Figure 20c). The current increases parabolically with the incident power, but the photocurrent does not reach saturation (inset to Figure 20c). Responsivity was calculated to be 5 mA/W. The rise and decay time were 0.35 ms and 0.25 ms, respectively, demonstrated a fast response behavior in the photodetector (Figure 20d). In addition, the EQE of the perovskites MWs-based device is twice as high as nanoparticles-based one, which demonstrated the morphological properties could play an essential role in photodetection.

Zhu et al. [70] constructed a FET based on MAPbI_3_ MWs to study photoelectronic characteristics. The schematic diagram of FET is shown in Figure 21a. SEM image of the FET based on MAPbI_3_ microwires was shown in the inset of Figure 21a. The on–off ratio was calculated to be 4.02 × 10^3^ (Figure 21b). A stable and reproducible I–t curve demonstrated the fast response and stability of the MAPbI_3_ MWs-based device (Figure 21c). Good performance should be beneficial to the carrier charge transport caused by the oriented alignment of the microwires and narrow electrode channel distance of about 20 mm. In addition, a stable and available ohmic contact make contributions as well. The rise and decay time was 0.2 ms and 0.25 ms, respectively (Figure 21d). The relatively low responsivity of 0.3 A W^−1^ may be caused by low efficient electron transfer during the light absorption.

Xiao et al. [71] fabricated a phototransistor based on MAPbI_3_ NWs with well-defined facets and smooth surfaces. The SEM image of the device was shown in Figure 22a. Under illumination, the I–V_sd_ curves are nonlinear, and the current reaches saturation at high V_sd_ (Figure 22b), which indicates a diode contact barrier. At high V_sd_, the saturation region exhibits a linear laser power-photocurrent curve (Figure 22c). The responsivity R was calculated to be 0.11 AW^−1^. A gain was estimated to be 0.25. In addition, they demonstrated that the photocurrent becomes stronger as the excitation energy close to the bandgap due to the strong light coupling, which provides new insights to the carrier transport and dynamics.

Spina et al. [72] fabricated a phototransistor based on MAPbI_3_ NWs/monolayer graphene to improve the performance of the pure MAPbI_3_-based photodetector by using high carrier mobility in graphene. Schematic diagram was shown in Figure 23a. In this device, positive charge carriers were injected into the graphene and negative charges accumulated in the NWs. The accumulating negative charges serve as an additional light tunable gate, further reducing the Fermi energy of graphene (Figure 23b). The key role of MAPbI_3_ NWs in the photon-induced carrier generation can be revealed by spectral sensitivity (Figure 23c). The responsivity reaches up to 2.6 × 10^6^ A W^−1^ (Figure 23e). The rise and decay time were 55 s and 75 s, respectively (Figure 23f). As the size of the device is reduced by 5-fold, the photoresponse is increased by about 10-fold. It should be contributed to the more effective collection of the photogenerated charge carriers.

Chen et al. [73] fabricated a phototransistor based on the C8BTBT/CsPbI_3_ nanorod heterojunction. Figure 24a illustrated the schematic of the phototransistor. The I_D_–V_D_ curve of the phototransistor exhibit both linear and saturation regions (Figure 24b,c). The transfer curves (I_D_–V_G_) as shown in Figure 24d show a typical p-type semiconductor behavior. The photocurrent shows that the larger gate voltage applied, the faster photocurrent increased (Figure 24e). The responsivity of 4.3 × 10^3^ A W^−1^ was obtained, as shown in Figure 24f. The on/off ratio was calculated to be 2.2 × 10^6^ as shown in Figure 24g. More importantly, due to the high stabilities of the two materials and device structure, the hybrid phototransistors possessed long-term stabilities under ambient conditions.

Meng et al. fabricated a phototransistor based on CsPbX_3_ (X = Cl, Br, or I) NWs with a uniform diameter of ~150 nm. As shown in Figure 25, these devices exhibit high performance with the responsivity exceeding 4489 A/W and detectivity over 7.9 × 10^12^ Jones. The response times are found to be less than 50 ms. The excellent performance can be attributed to the reduced defect concentration in CsPbX_3_ NWs as well as the field-effect transistors (FET) with superior hole field-effect mobility of 3.05 cm^2^/(V s) [74].

Yang et al. constructed a phototransistor based on CsPbI_3_ nanorods. The schematic diagram of the device was shown in Figure 26a. The linear and symmetric I–V curves demonstrated that the contact was ohmic, as shown in Figure 26b. The as-fabricated device exhibited a totally excellent performance, such as high responsivity of 2.92 × 10^3^ A·W^−1^, large EQE of 0.9 × 10^6^%, fast response time of 0.05 ms, and a high detectivity of 5.17 × 10^13^ Jones (Figure 26c–f). The excellent performance is mainly due to the following two reasons. First, high absorption coefficient, low recombination of charge carriers and low density of defects of CsPbI_3_ nanorods generate strong photoelectric effect. Second, the high-quality nanorod provides a smooth and short path for carrier transfer, and significantly improves the response speed [75].

Du et al. developed a phototransistor based on CsPbI_3_ nanotubes, which can be stable for more than two months under air conditions. The schematic diagram was shown in Figure 27a. The phototransistor exhibited an excellent performance with an EQE, detectivity, photoresponsivity and response time of 5.65 × 10^5^%, 9.99 × 10^13^ Jones, 1.84 × 10^3^ A W^−1^ and 3.78 ms/359 ms, respectively (Figure 27b–d). It is comparable to the best of all inorganic perovskite photodetectors, which is mainly attributed to the enhanced light absorption resulting from the light trapping effect within the tube cavity [76].

#### 2.2.3. Phototransistor Based on Other Nanostructured Perovskites

Kwak et al. [77] constructed a graphene/CsPbBr_3-x_I_x_ NCs-based photodetector with high performance. Figure 28a illustrated the schematic of the hybrid photodetector. A responsivity as high as 8.2 × 10^8^ A W^−1^ and detectivity of 2.4 × 10^16^ Jones were achieved (Figure 28b). The high performance of the phototransistor based on graphene/CsPbBr_3-x_I_x_ NCs can be attributed to the fast carrier transport of graphene and strong light absorption of perovskite NCs. The photocurrent increased obviously with the increasing of incident power and showed a good on/off switching behavior (Figure 28c). The rise and decay time as shown in Figure 28d were calculated to be 0.81 s and 3.65 s, respectively.

Surendran et al. [78] demonstrated another graphene/CsPbBr_x_I_3-x_ NCs-based phototransistor. Figure 29a shows the schematic diagram of the hybrid phototransistor. The PL peak and absorption edge can be redshifted to 650 nm (Figure 29b). The obviously decreased PL lifetime of hybrid structure (Figure 29c) indicates that massive charge extracted from perovskite layer and injected into graphene, and thus improving the photodetector performance (Figure 29d). The responsivity of 1.12 × 10^5^ A/W and the detectivity of 1.17 × 10^11^ Jones were achieved in this device (Figure 29e). The rise and decay times were 273.6 ms and 2.26 s, respectively, with excellent reproducibility (Figure 29f). Meanwhile, a large photoconductive gain of 9.32 × 10^10^ further demonstrated the potential for detecting extremely low power light. The device exhibits a significant improvement in stability with the photocurrent retention of ~82% after 37 h.

Wu et al. [44] fabricated a phototransistor based on CsPbI_3-*x*_Br*_x_* quantum dots (QDs)/monolayer MoS_2_ heterostructure with high-performance and low-cost. The schematic diagram of the phototransistor was shown in Figure 30a. The TEM image shown in Figure 30a inset demonstrated a cubic shape of the CsPbI_3-*x*_Br*_x_* quantum dots. The linear and symmetric *I*_D_–*V*_D_ curves indicate a Schottky barriers at the contact interface. As shown in Figure 30b,c, the on/off ratio exceeds 10^4^, the photoresponsivity reach up to 7.7 × 10^4^ A W^−1^, the specific detectivity of 5.6 × 10^11^ Jones and an ultrahigh EQE over 10^7^%. It should be noted that both R and D* decreased exponentially with increasing incident power because of high recombination and scattering. Stable and reproducible showed a good on/off photoswitching characteristic (Figure 30d). The rise and decay times were 0.59 s and 0.32 s, respectively (Figure 30e).

Zou et al. [79] report a phototransistor based on CsPbI_3_ QD/DPP-DTT heterojunction. The DPP-DTT polymer with narrow bandgap and high carrier mobility was chosen to improve the detection range and performance. The schematic was shown in Figure 31a. Figure 31b showed the intersecting surface. The phototransistor exhibits broadband detection from 350 to 940 nm by combining UV–vis absorption of CsPbI_3_ QDs and NIR absorption of DPPDTT. A high responsivity of 110 A W^−1^ (Figure 31c) and a specific detectivity of 2.9 × 10^13^ Jones (Figure 31d) were achieved due to the heterojunction strategy and gate modulation. The on/off ratio of 6 × 10^3^ (Figure 31e) indicated good photoswitching characteristics. In addition, the responsivity can be maintained 80% after one month demonstrated an excellent stability of the device.

### 2.3. Photodiode Based on Nanostructured Perovskites

Since most of the photodiode is based on thin film that is constructed by nanostructured perovskites, this part is not divided into different dimensions here. Easy to combine with other materials makes polycrystalline film a perfect choice for making photodiodes. The first perovskite photodiode using polycrystalline film as the active layer was reported by Yang et al. The device with the structure of PEDOT: PSS/MAPbI_3−x_Cl_x_/ PCBM/Al was adopted. PEDOT: PSS served as hole-transporting layer (HTL), PCBM as electron-transporting layer (ETL), and water/alcohol-soluble conjugated polymer serve as hole blocking layer. 2,9-dimethyl-4,7-diphenyl-1,10-phenanthroline (BCP) and poly[(9,9-bis(3′-(N,N-dimethylamino)propyl)-2,7-fluorene)-alt-2,7-(9,9-dioctylfluorene)] (PFN) were added to reduce the dark current density under reverse bias (Figure 32a,b). The photodetectors exhibit a high detectivity, a fast photoresponse and a linear dynamic range. (Figure 32c,d) [80] The performance is even better than commercial Si-based photodetectors. The HTL was replaced with cross-linkable N4,N4-Bis(4-(6-([3-ethyloxetan-3-yl]methoxy)hexyl)phenyl)-N4,N4-diphe nylbiphenyl-4,4-diamine (OTPD) layer to reduce the dark current (Figure 32e,f). Meanwhile, PCBM/C60 double layer was employed as the hole blocking layer and C60 can passivate most of the charge traps. As a result, the dark current was suppressed to 9.1 × 10^−9^ A cm^−2^ at −2 V, which was much lower than the former detector using PEDOT: PSS as HTL. In addition, as shown in Figure 32g,h, the photodetector exhibits an ultrahigh detectivity of 7.4 × 10^12^ Jones, a large LDR of 94 dB and fast response time (120 ns) [81].

Lin et al. also chose a thick PCBM/C60 interlayer as hole blocking layer to coat the perovskite homojunction and the dark current was reduced to 5 × 10^−10^ A cm^−2^ under −0.5 V [82]. Meanwhile, as a low shunt capacitor, the interlayer plays a key role in improving the diode temporal response. The interlayer also provides electro-optical control of the spectral response. Sutherland et al. fabricate a photodiode with a structure of Au/Spiro/MAPbI_3_/TiO_2_/FTO [83]. Addition of Al_2_O_3_ and PCBM layer between TiO_2_ and MAPbI_3_ can effectively reduce the dark current and improve the responsivity, which should be consistent with the posited passivation of charge-trapping recombination centers at the interface (Figure 33a,b). High sensitivity throughout the visible and into the near-infrared region. As shown in Figure 33c,d, both the peak responsivity (0.395 A W^−1^) and specific detectivity (10^12^ Jones) can be comparable to those of commercial silicon photodetectors [84,85]. Furthermore, the photodetector exhibits bias-independent responsivity and stable photocurrent after the detection of one billion laser pulses. These parameters indicate that perovskite materials have great potential in high-performance photodetectors. Dong et al. fabricated a photodiode with a structure of Ag/MoO_3_/4,4′-bis[(p-trichlorosilylpropylphenyl)phenylamino]-biphenyl(TPD-Si_2_)/MAPbI_3_. TPD-Si_2_ serves as the blocking layer. MoO_3_ is used for anode work function modification. PCBM/C60 passivation layers were cancelled to reserve surface trap states (Figure 33e,f). The large gain, which results from the hole traps caused by large concentration of Pb^2+^ cations in the perovskite film surface, makes the as-fabricated devices work as photoconductor rather than photodiode under illumination. The photodetector exhibits a broadband response ranging from the UV to the NIR, very high responsivity of 242 A W^−1^, a short response time and an excellent LDR of 85 dB (Figure 33g,h) [86].

In Table 2, we summarize some key parameters of the photoconductors based on nanostructured perovskites. It seems that 1D perovskite-based photodetectors are much more studied and often perform better than 2D perovskite-based photodetectors. This may be due to the relatively better crystal quality of 1D perovskites and better ohmic contact between 1D perovskites and electrodes.

## 3. Conclusions and Outlook

In summary, the research advancements of nanostructured perovskite-based photodetectors are reviewed. The performances of the detectors are influenced by the type of devices as well as the structures of nanostructured perovskites with different morphology. As a whole, photodetectors based on nanostructure perovskites perform well due to long carrier lifetime, great carrier mobility, and low carrier recombination result from few grain boundary and lower trap state density. Our conclusion is that nanostructured perovskites have great potential to be applied in low-voltage, low-cost, fast-response, high-detectivity, and ultra-highly integrated optoelectronic devices.

Despite great progress has been made on the perovskite photodetectors, there are still many challenges. First, both the photoconductors and phototransistors only achieve part of photodetection parameters enhancement. However, an ideal photodetector should improve performance including high responsivity, large detectivity, fast speed, etc. Second, perovskites easily degrade in air with the presence of oxygen and moisture. Thus, the instability of perovskite impedes the commercial use of the device. It is of great significance to improve the perovskite with new protection strategies though the heterojunction structures exhibit relatively better stability. In addition, some literatures only reported the best performance of the device, but ignored the average performance of the device, which would lead to a misleading effect on the industrialization direction, and also show that the authors have no confidence in the stability of the device. Finally, the toxicity of perovskite has cast a shadow over its application because of the use of lead in widely studied materials, such as MAPbI_3_, and CsPbBr_3_. Therefore, more efforts should be made to prepare environmental perovskite materials with non-toxic elements instead of Pb.

## Figures and Tables

**Figure 1 nanomaterials-11-01038-f001:**
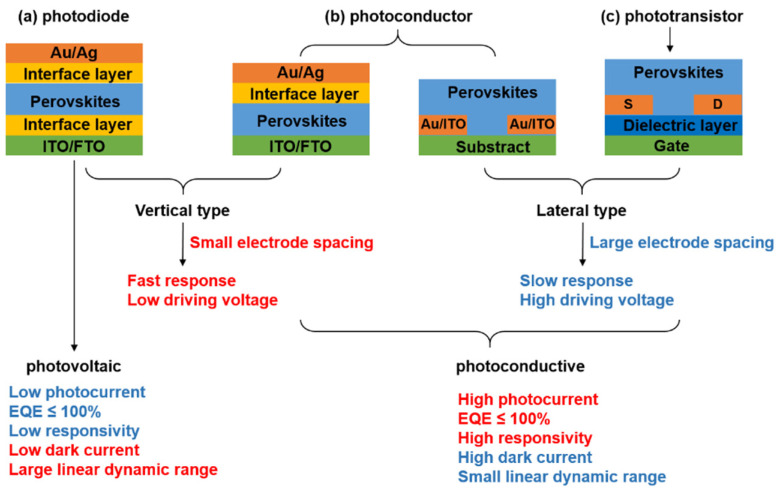
Schematic and characteristics of perovskite-based photodetectors. (**a**) photodiode, (**b**) photoconductor, and (**c**) phototransistor.

**Figure 2 nanomaterials-11-01038-f002:**
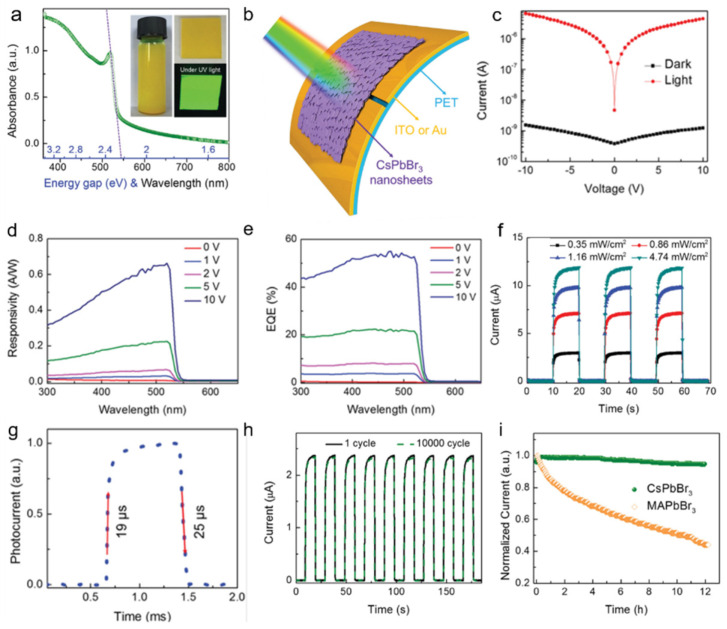
(**a**) UV–vis absorbance spectrum of CsPbBr_3_ nanosheets. (**b**) The schematic of a flexible photodetector device based on the CsPbBr_3_ nanosheet. (**c**) Logarithmic *I*–*V* characteristics of the photodetector in the dark and under irradiation with 442 nm light. (**d**) Responsivity and (**e**) EQE spectra of the photodetector. (**f**) *I*–*t* curves of the photodetector under 442 nm irradiation with different intensities. (**g**) Rise and decay times of the photodetector. (**h**) Photoresponse for the 1st and 10,000th bending-recovery cycle. (**i**) Excellent stability of photocurrent of CsPbBr_3_ photodetectors. Reprinted with permission from ref. [43]. Copyright 2016 Wiley-VCH.

**Figure 3 nanomaterials-11-01038-f003:**
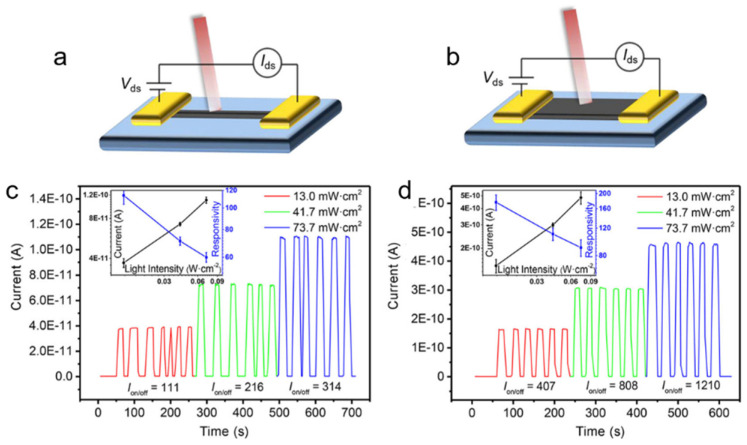
(**a**,**b**) Schematics of photodetectors based on an individual (**a**) nanowire crystal and (**b**) nanoplate crystal of MAPbI_3_. (**c**,**d**) Time response behavior of the photodetectors based on wire and platelet crystals, respectively. Reprinted with permission from ref. [53]. Copyright 2016 Wiley-VCH.

**Figure 4 nanomaterials-11-01038-f004:**
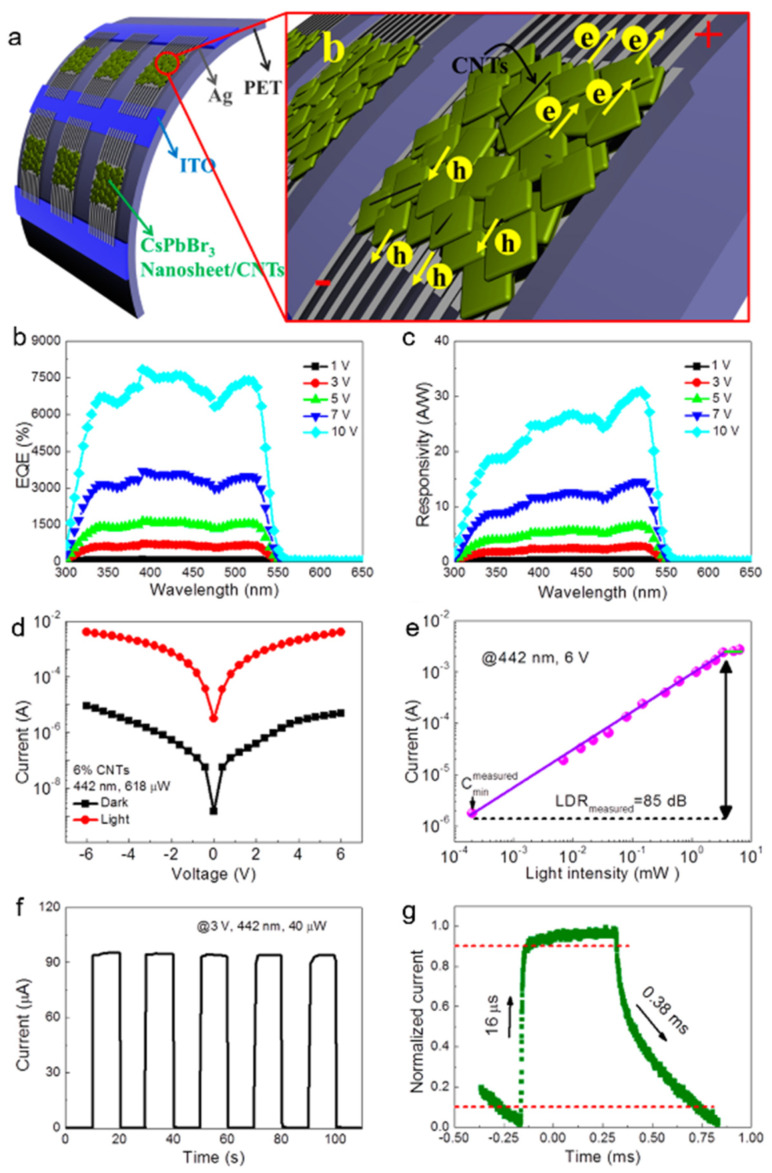
(**a**) Schematic illustration of the CsPbBr_3_ nanosheet/ carbon nanotube (CNT)-based photodetector. (**b**) EQE and (**c**) responsivity spectra under different biases. (**d**) Logarithmic I–V curves under dark and 442 nm light illumination. (**e**) Photocurrent versus light intensity of the photodetector. (**f**) I–t curve of the photodetector. (**g**) Rise and decay time of the device. Reprinted with permission from ref. [55]. Copyright 2017 American Chemical Society.

**Figure 5 nanomaterials-11-01038-f005:**
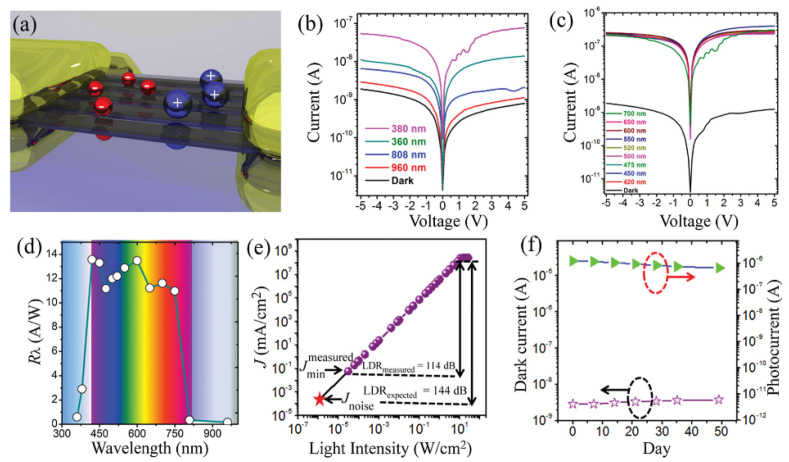
(**a**) Device structure of the MAPbI_3_ microwire (MW) array-based photodetector. (**b**,**c**) I–V curves of a typical photodetector. (**d**) Photoresponsivity and photodetectivity of the MAPbI_3_ MW array-based photodetector. (**e**) Photocurrent versus light intensity of the device under 550 nm light illumination. (**f**) Variation of dark current/photocurrent of MAPbI_3_ MW arrays-based photodetector in ambient air. Reprinted with permission from ref. [56]. Copyright 2016 Wiley-VCH.

**Figure 6 nanomaterials-11-01038-f006:**
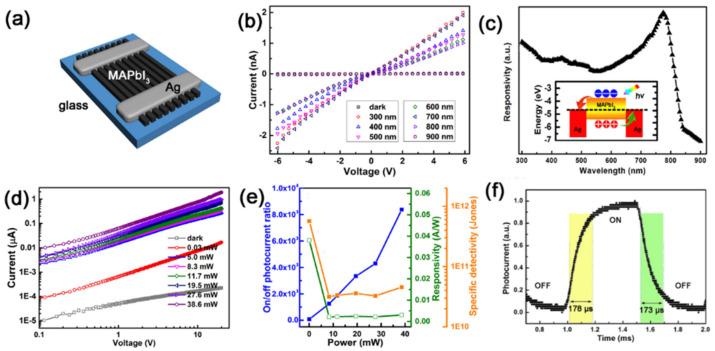
(**a**) The schematic diagram of the MAPbI_3_ MWs-based photodetector. (**b**) I–V curves of the photodetector. (**c**) Wavelength-dependent responsivity of the photodetector. The inset is the energy band diagrams of the structure. (**d**) I–V curve of the photodetector with various light illumination power (650 nm). (**e**) On/off ratio, responsivity, and specific detectivity of the photodetectors. (**f**) Rise and decay time of the device. Reprinted with permission from ref. [57]. Copyright 2019 Elsevier.

**Figure 7 nanomaterials-11-01038-f007:**
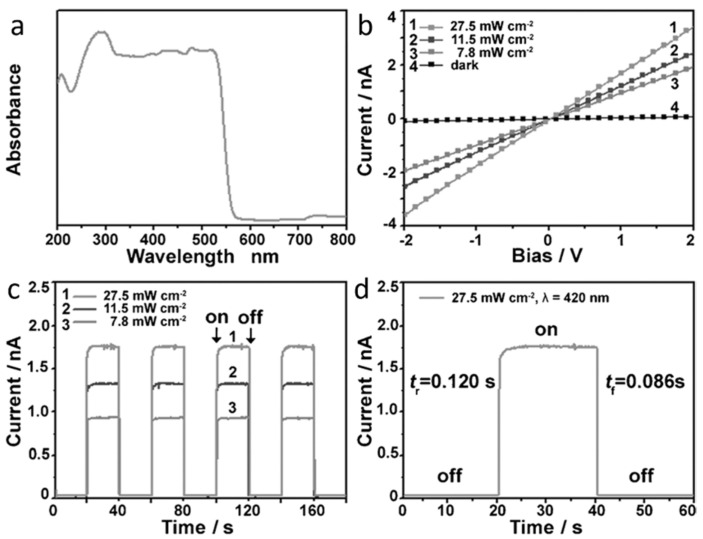
(**a**) UV/Vis absorption spectrum. (**b**) Typical I–V curves. (**c**,**d**) Photoresponse of the porous CH_3_NH_3_PbBr_3_ nanowires (NWs). Reprinted with permission from ref. [58]. Copyright 2015 Wiley-VCH.

**Figure 8 nanomaterials-11-01038-f008:**
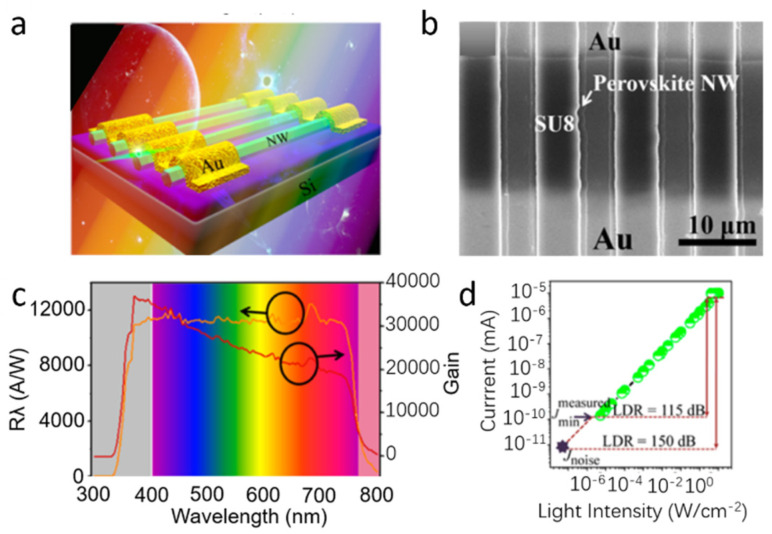
(**a**) Schematic of transistor based on the MAPbI_3_ NWs. (**b**) SEM (scanning electron microscope) images of the phototransistor. (**c**) Responsivity and gain under different wavelength. (**d**) Dynamic response of the phototransistor. Reprinted with permission from ref. [59]. Copyright 2017 American Chemical Society.

**Figure 9 nanomaterials-11-01038-f009:**
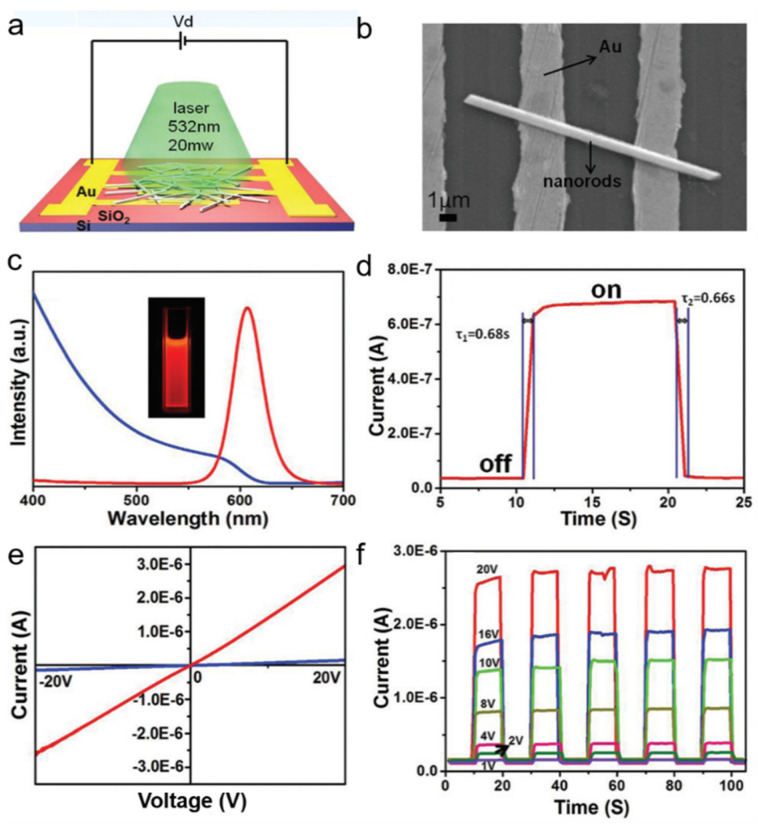
(**a**) The schematic of CsPb(Br/I)_3_ nanorods-based photodetector; (**b**) SEM image of the device. (**c**) photoluminescence (PL) and absorption spectrum of the nanorods. (**d**) Rise and decay time of the photodetector. (**e**) I–V curve of the photodetector. (**f**) I–t curve as a function of bias voltages. Reprinted with permission from ref. [60]. Copyright 2016 Royal Society of Chemistry.

**Figure 10 nanomaterials-11-01038-f010:**
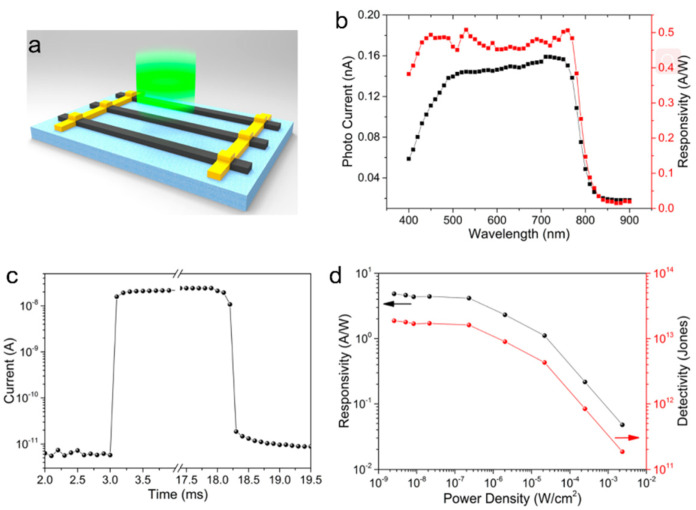
(**a**) Schematic of device structure of device. (**b**) Wavelength-dependent photocurrent and responsivity. (**c**) Transient response (current–time curve). (**d**) Light intensity dependent responsivity and detectivity. Copyright 2016, American Chemical Society [61].

**Figure 11 nanomaterials-11-01038-f011:**
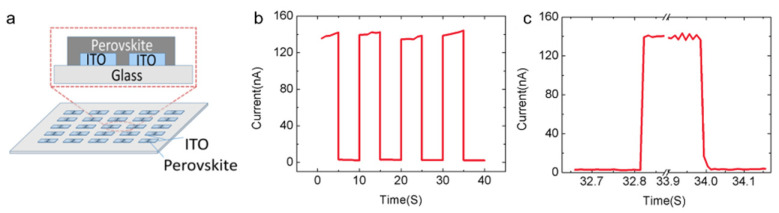
(**a**) Schematic of the photodetector. (**b**) Current-time curve of the MW-based photodetector. (**c**) Photoresponse time of the MW-based photodetector. Reprinted with permission from ref. [62]. Copyright 2019 American Chemical Society.

**Figure 12 nanomaterials-11-01038-f012:**
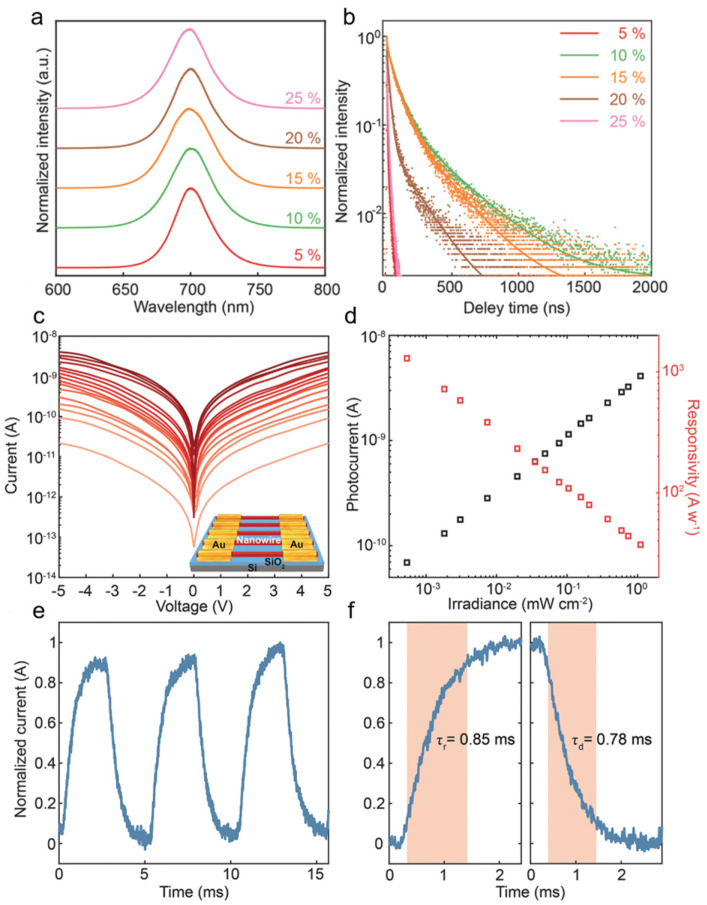
(**a**) Normalized PL emission, and (**b**) time-resolved PL spectra of CsPbI_3_ NWs. (**c**) The typical I–V curves of α-CsPbI_3_ nanowire arrays, inset is the schematic illustration of device. (**d**) Photocurrent and responsivity of photodetector. (**e**) I–t response and (**f**) photoresponse time of photodetectors. Reprinted with permission from ref. [42]. Copyright 2019 Wiley-VCH.

**Figure 13 nanomaterials-11-01038-f013:**
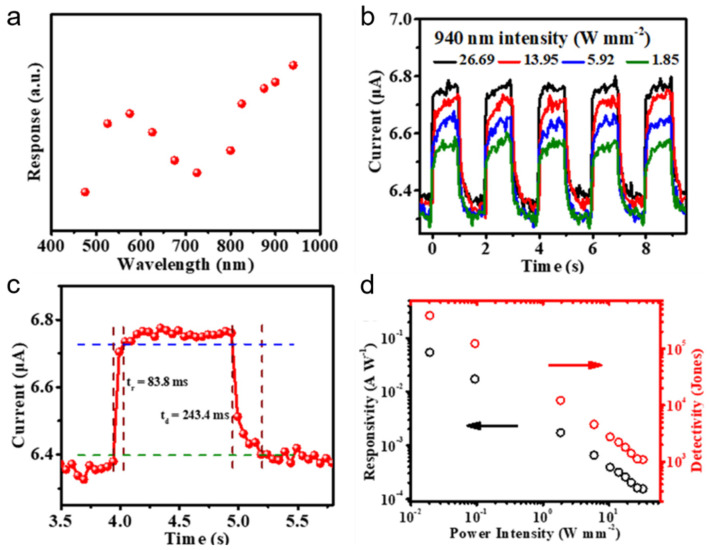
Near-infrared photodetection performance of CsSnI_3_ NW array-based photodetector. (**a**) Spectral response of the photodetector illuminated from 475 to 940 nm. (**b**) Time-response curves of photodetector. (**c**) Rise and decay time constants. (**d**) Responsivity and detectivity of the photodetector. Reprinted with permission from ref. [63]. Copyright 2019 Scientific Publishers of India.

**Figure 14 nanomaterials-11-01038-f014:**
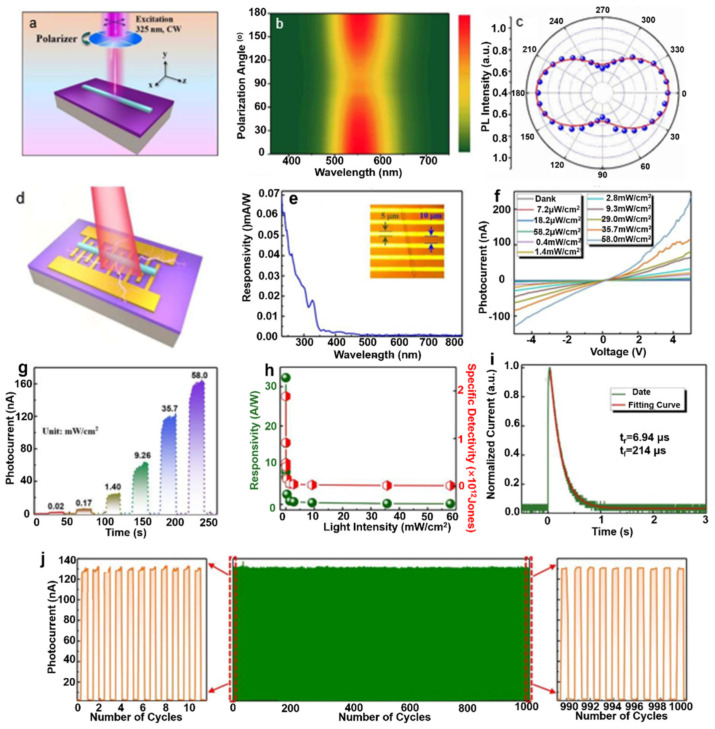
(**a**) Schematic diagram of polarization PL measurements. (**b**) PL spectra vs. incident light polarization. The color represents the emission intensity. (**c**) Polarization-dependent PL intensity of CsCu_2_I_3_ NWs. (**d**) Schematic diagram of the CsCu_2_I_3_ NW-based photodetector. (**e**) Responsivity for different incident wavelength. Inset is the optical microscope image of the photodetector. (**f**) I–V and (**g**) I–t curves of the photodetector with different incident power. (**h**) Responsivity and specific detectivity with different incident power. (**i**) Rise and decay time of the photodetector. (**j**) Photoresponse maintained well after 1000 bending cycles. Reprinted with permission from ref. [64]. Copyright 2020 Royal Society of Chemistry.

**Figure 15 nanomaterials-11-01038-f015:**
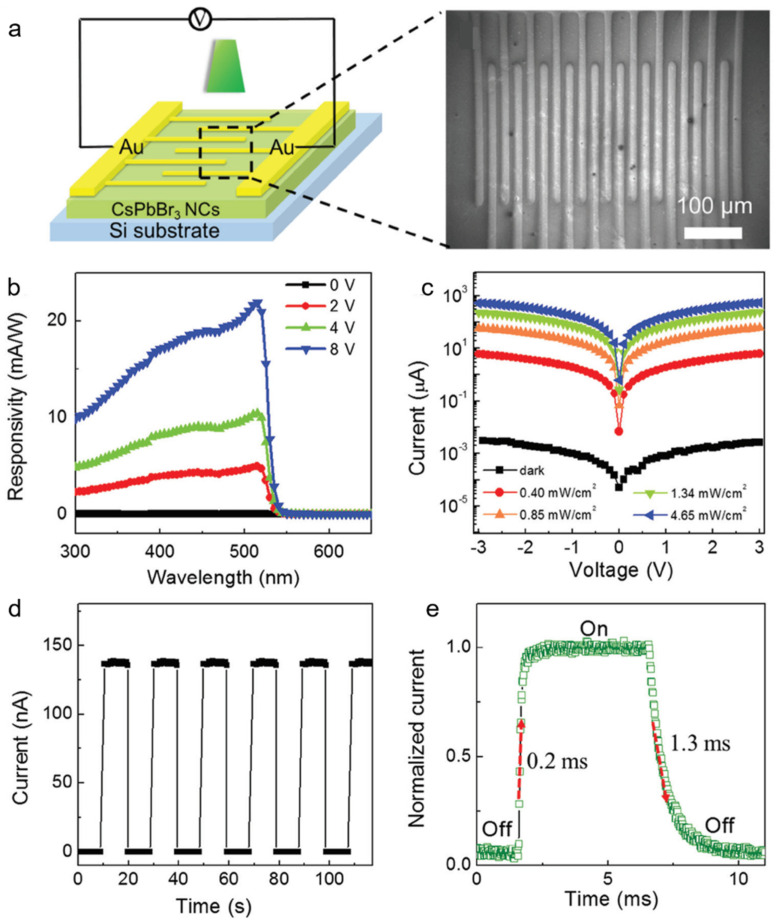
(**a**) Schematic and microscope image of CsPbBr_3_ NC-based photodetector. (**b**) Responsivity of the photodetector. (**c**) I–V curves of photodetector with different light irradiation intensity. (**d**) I-t curve of photodetector. (**e**) Rise and decay time of the device. Reprinted with permission from ref. [65]. Copyright 2016 Wiley-VCH.

**Figure 16 nanomaterials-11-01038-f016:**
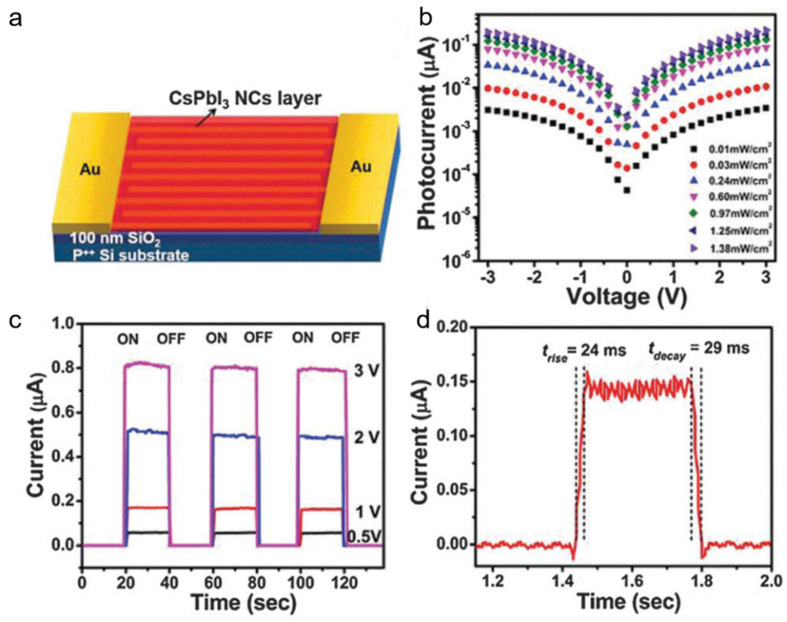
(**a**) Schematic diagram of the CsPbI_3_ nanocrystals (NCs)-based photodetector. (**b**) I–V curve of photodetector with incident light intensity. (**c**) I–t curve as a function of applied bias. (**d**) Rise and decay time of the device. Reprinted with permission from ref. [52]. Copyright 2016 Royal Society of Chemistry.

**Figure 17 nanomaterials-11-01038-f017:**
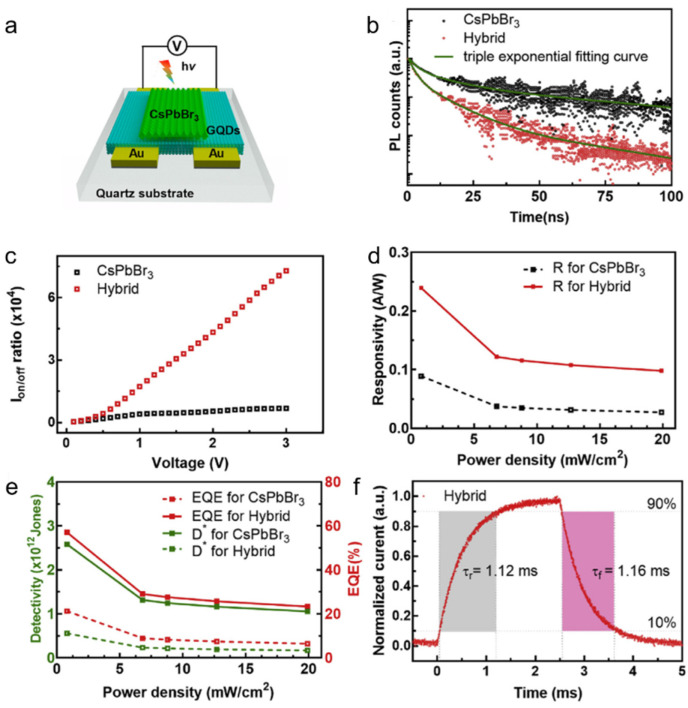
(**a**) Schematic diagram of the graphene quantum dots (GQDs)/CsPbBr_3_-based photodetector. (**b**) Time-resolved PL decay transients for the pure and hybrid structures. (**c**) On/off ratio, (**d**) Photoresponsivity, (**e**) specific detectivity (D*), and EQE of the pure CsPbBr_3_-based and hybrid photodetectors. (**f**) Rise and decay time of the hybrid photodetector. Reprinted with permission from ref. [66]. Copyright 2020 Elsevier.

**Figure 18 nanomaterials-11-01038-f018:**
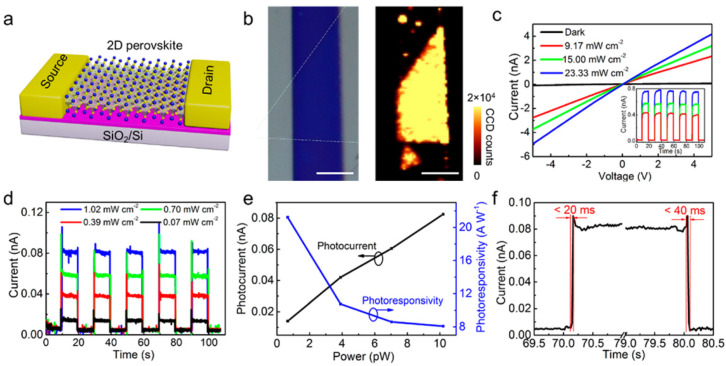
(**a**) Schematic diagram of a phototransistor based on 2D MAPbX_3_ nanosheet. (**b**) Picture (left) and PL mapping image (right) of field-effect transistor (FET). (**c**) I–V curves of the FET. Inset: I–t curve under different power with a voltage bias of 1 V. (**d**) I–t curve of the FET under the different power. (**e**) Photocurrent and responsivity as a function of incident power. (**f**) Rise and decay times of the FET are within 20 and 40 ms, respectively. Reprinted with permission from ref. [67]. Copyright 2016 American Chemical Society.

**Figure 19 nanomaterials-11-01038-f019:**
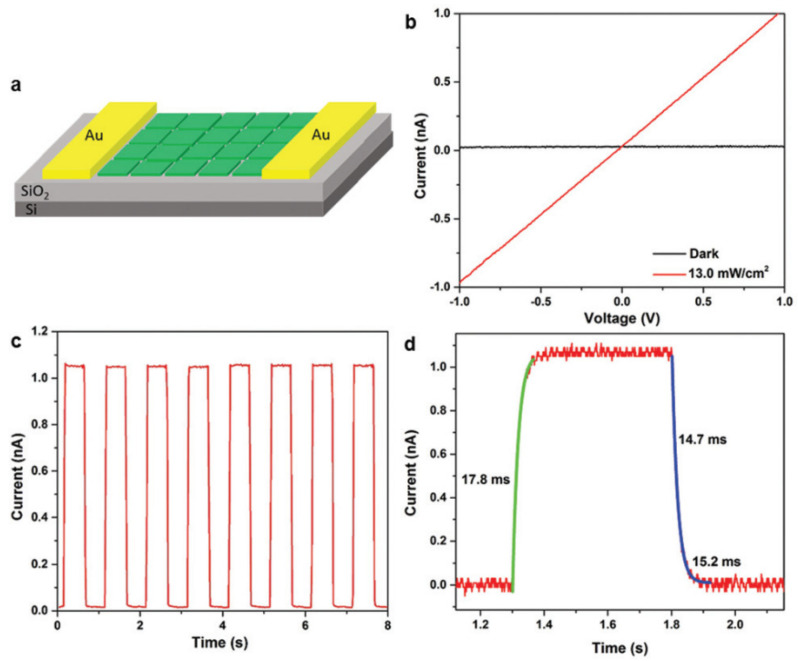
(**a**) Schematic diagram of the phototransistor based on CsPbBr_3_ nanosheets. (**b**) I–V curves of the photodetector. (**c**) I–t curve of the photodetector. (**d**) Rise and decay times of the photodetector. Reprinted with permission from ref. [68]. Copyright 2016 Royal Society of Chemistry.

**Figure 20 nanomaterials-11-01038-f020:**
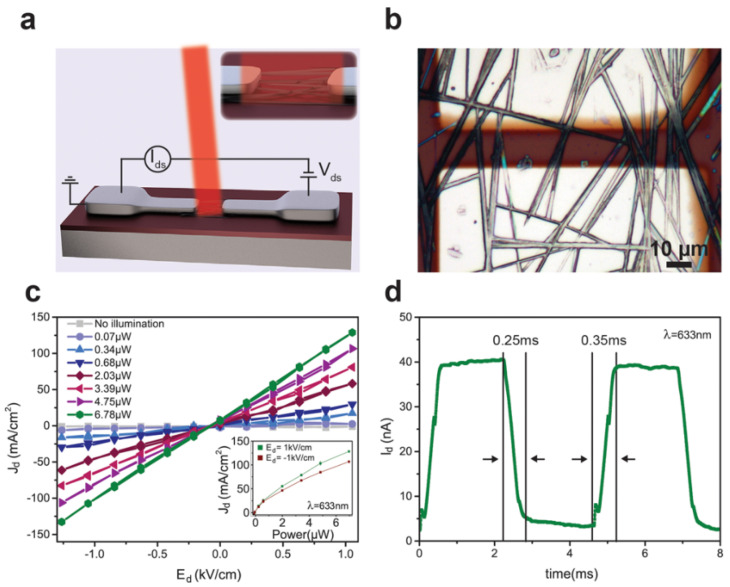
(**a**) Schematic diagram of the MAPbI_3_ NWs-based FET and I–V photocurrent measurements. (**b**) Microscopy image of the FET based on MAPbI_3_ NWs. (**c**) Dark and laser illuminated I–V curves as a function of laser power. (**d**) Rise and decay time of the FET. Reprinted with permission from ref. [69]. Copyright 2014 American Chemical Society.

**Figure 21 nanomaterials-11-01038-f021:**
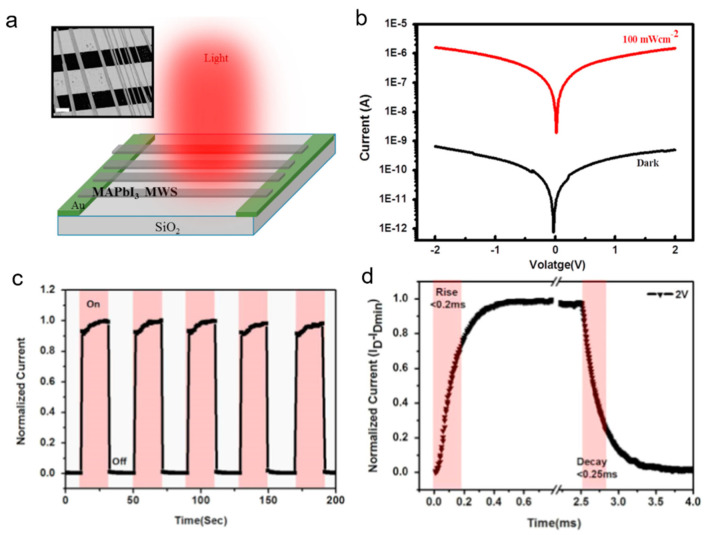
(**a**) Schematic diagram of the FET based on MAPbI_3_ MWs, inset: SEM image of the FET. (**b**) I–V curve of the photodetector under dark and illumination at 100 mWcm^−2^. (**c**) I–t curves of the photodetector. (**d**) Rise and decay time were 0.2 ms and 0.25 ms, respectively. Reprinted with permission from ref. [70]. Copyright 2016, Elsevier.

**Figure 22 nanomaterials-11-01038-f022:**
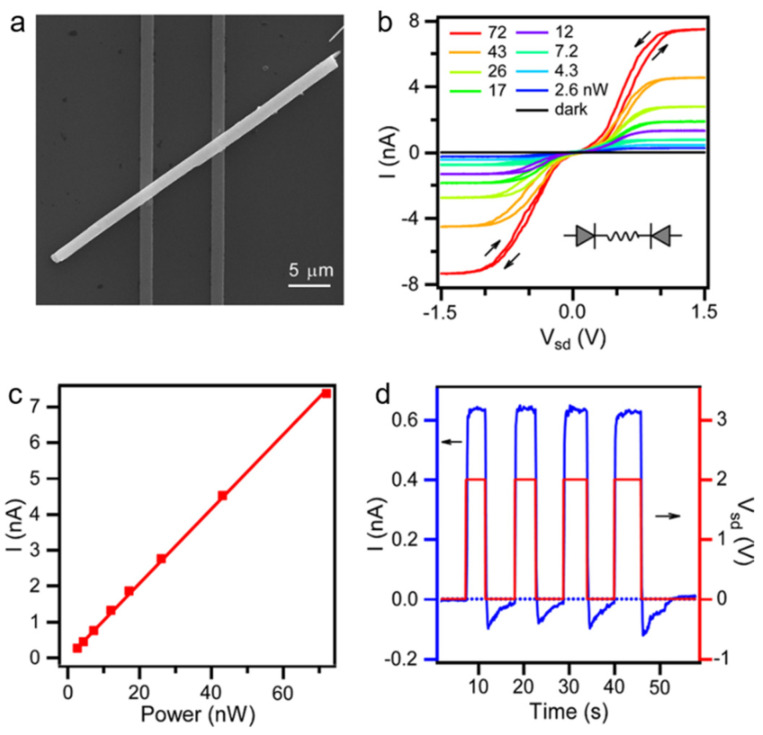
(**a**) SEM image of the MAPbI_3_ NM-based photodetector. (**b**) I–V_sd_ curves of the device. (**c**) Laser-power-dependent photocurrent in the saturation region at high V_sd_. (**d**) I–t curves of the photodetector. Reprinted with permission from ref. [71]. Copyright 2016 American Chemical Society.

**Figure 23 nanomaterials-11-01038-f023:**
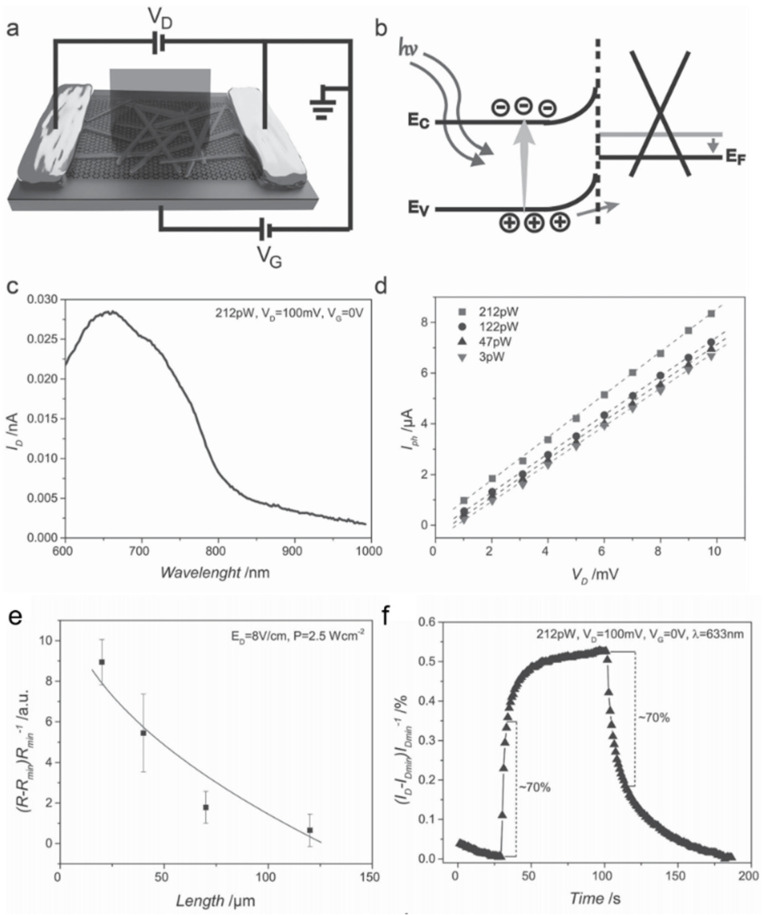
(**a**) Schematic diagram of the phototransistor based on MAPbI_3_ NWs/monolayer graphene. (**b**) The band diagram of the MAPbI_3_ NW/graphene heterojunction. (**c**) Wavelength-dependent photocurrent. (**d**) I–V curves of the photodetector. (**e**) The responsivity as a function of the device length. (**f**) Time response of the phototransistor. Reprinted with permission from ref. [72]. Copyright 2015 Wiley-VCH.

**Figure 24 nanomaterials-11-01038-f024:**
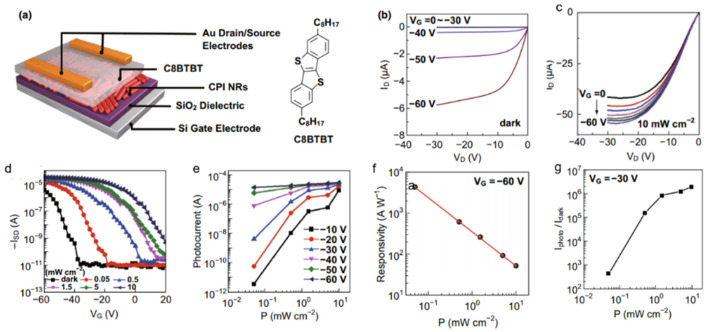
(**a**) Schematic diagram of phototransistor based on C8BTBT/CsPbI_3_ nanorod heterojunction. ID–VD transistor characteristics in the dark state (**b**) and under a white-light illumination (**c**). (**d**) Transfer characteristics (V_D_ = −30 V) under different incident power. (**e**) Photocurrent as a function of the incident power under different gate voltages. (**f**) Responsivity as a function of incident power. (**g**) On/off ratio as a function of incident power. Reprinted with permission from ref. [73]. Copyright 2018 Springer Singapore.

**Figure 25 nanomaterials-11-01038-f025:**
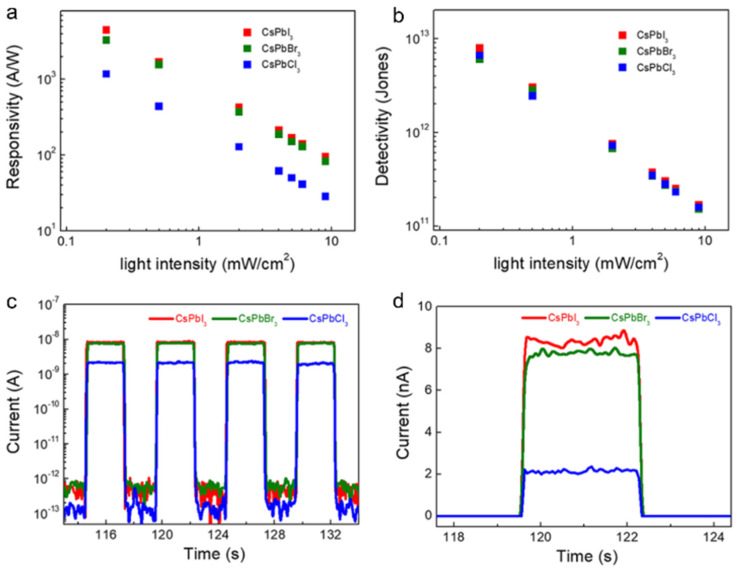
(**a**,**b**) Responsivity and detectivity vs. light intensity. (**c**,**d**) I–t curve under illumination (9 mW/cm^2^) and a bias voltage of 5 V. Reprinted with permission from ref. [74]. Copyright 2019 American Chemical Society.

**Figure 26 nanomaterials-11-01038-f026:**
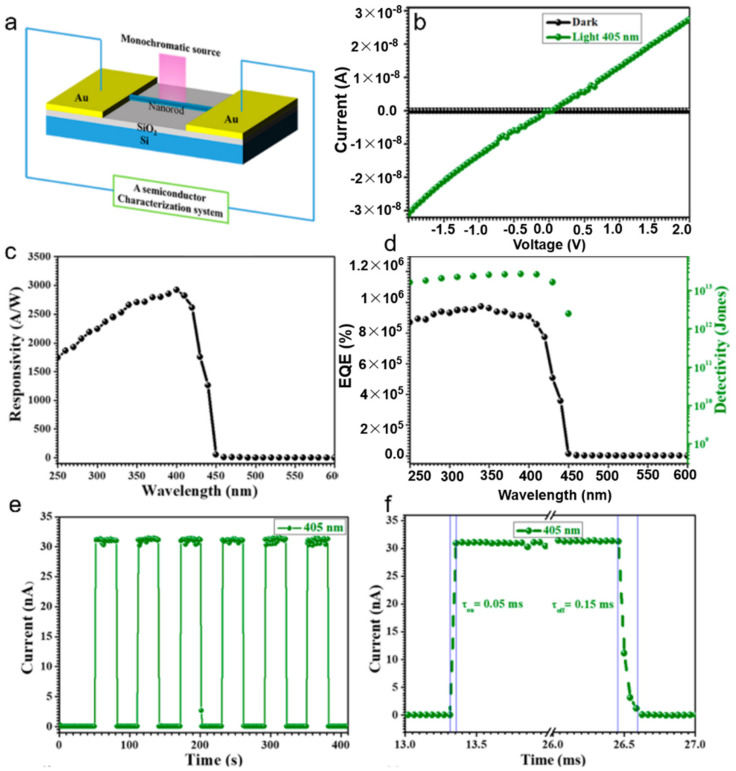
(**a**) Schematic diagram of the as-fabricated phototransistor based on a single CsPbI_3_ nanorod. (**b**) I–V characteristics under dark and 405 nm irradiation. (**c**,**d**) Representative, EQE and Detectivity vs. wavelength. (**e**,**f**) I–t curve and response time of the photodetector. Reprinted with permission from ref. [75]. Copyright 2018 American Chemical Society.

**Figure 27 nanomaterials-11-01038-f027:**
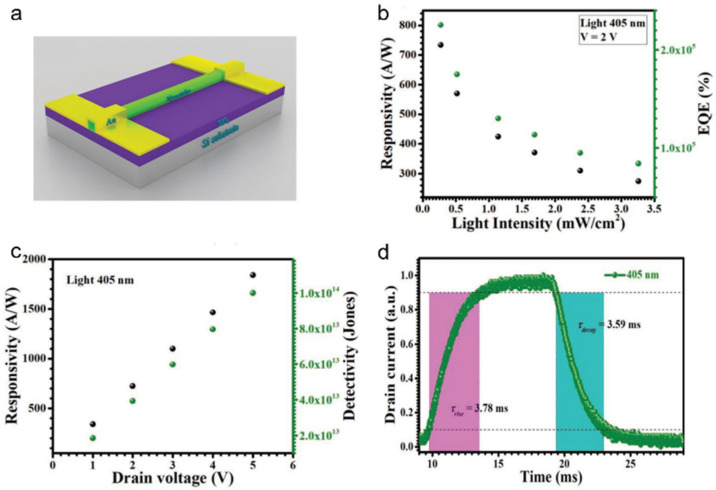
(**a**) The schematic diagram for the phototransistor based on CsPbI_3_ nanotubes. (**b**) Responsivity and EQE vs. incident intensity. (**c**) Responsivity and Detectivity vs. drain voltage. (**d**) Response time of the device. Reprinted with permission from ref. [76]. Copyright 2019 Wiley-VCH.

**Figure 28 nanomaterials-11-01038-f028:**
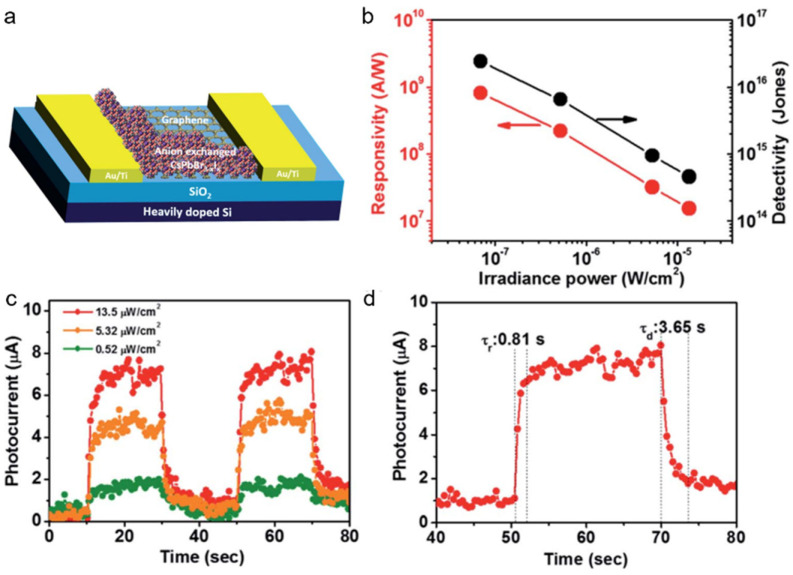
(**a**) Schematic of graphene/CsPbBr_3-x_I_x_ NCs-based photodetector. (**b**) Responsivity and detectivity of the phototransistor. (**c**) I–t curve under different incident power. (**d**) Response time of the photodetector. Reprinted with permission from ref. [77]. Copyright 2015 Wiley-VCH.

**Figure 29 nanomaterials-11-01038-f029:**
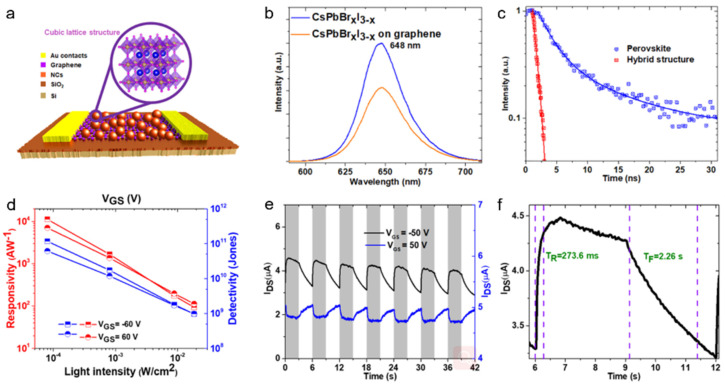
(**a**) Schematic diagram of the hybrid phototransistor, inset: the crystal structure of CsPbBr_x_I_3-x_ NCs. (**b**) PL spectra of the hybrid structure. (**c**) A decrease of time-resolved PL in hybrid structure. (**d**) Responsivity and detectivity as a function of incident power. (**e**) I–t curve under different gate biases. (**f**) Response time of the phototransistor. Reprinted with permission from ref. [78]. Copyright 2019 Wiley-VCH.

**Figure 30 nanomaterials-11-01038-f030:**
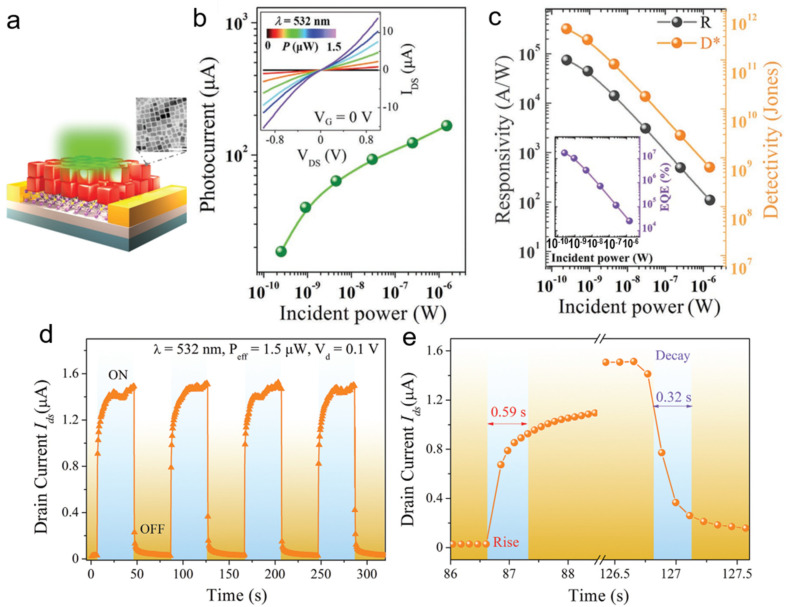
(**a**) Schematic of phototransistor based on CsPbI_3-*x*_Br*_x_* quantum dots/monolayer MoS_2_ heterostructure, inset: TEM image of the CsPbI_3-*x*_Br*_x_* quantum dots. (**b**) Photocurrent of the phototransistor. The inset shows the I_D_–V_D_ curves under different incident powers. (**c**) Photoresponsivity and specific detectivity of the phototransistor, inset: the EQE as a function of incident power. (**d**) I–t curve demonstrated a stable and reproducible photoswitch. (**e**) Response time of the device. Reprinted with permission from ref. [44]. Copyright 2018 Wiley-VCH.

**Figure 31 nanomaterials-11-01038-f031:**
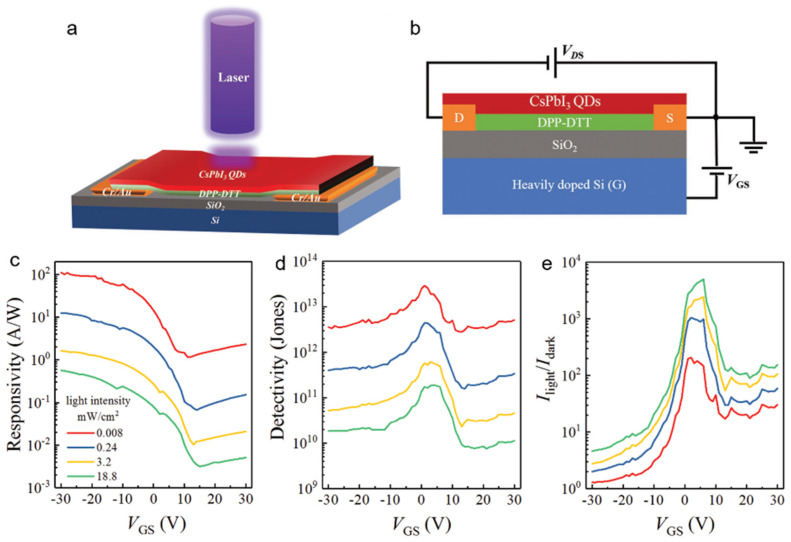
(**a**) The schematic of the phototransistor based on CsPbI_3_ QD/DPP-DTT heterojunction. (**b**) The intersecting surface of the device. Gate-dependent (**c**) responsivity, (**d**) specific detectivity, and (**e**) light–dark current ratios as a function of incident power. Reprinted with permission from ref. [79]. Copyright 2018 Wiley-VCH.

**Figure 32 nanomaterials-11-01038-f032:**
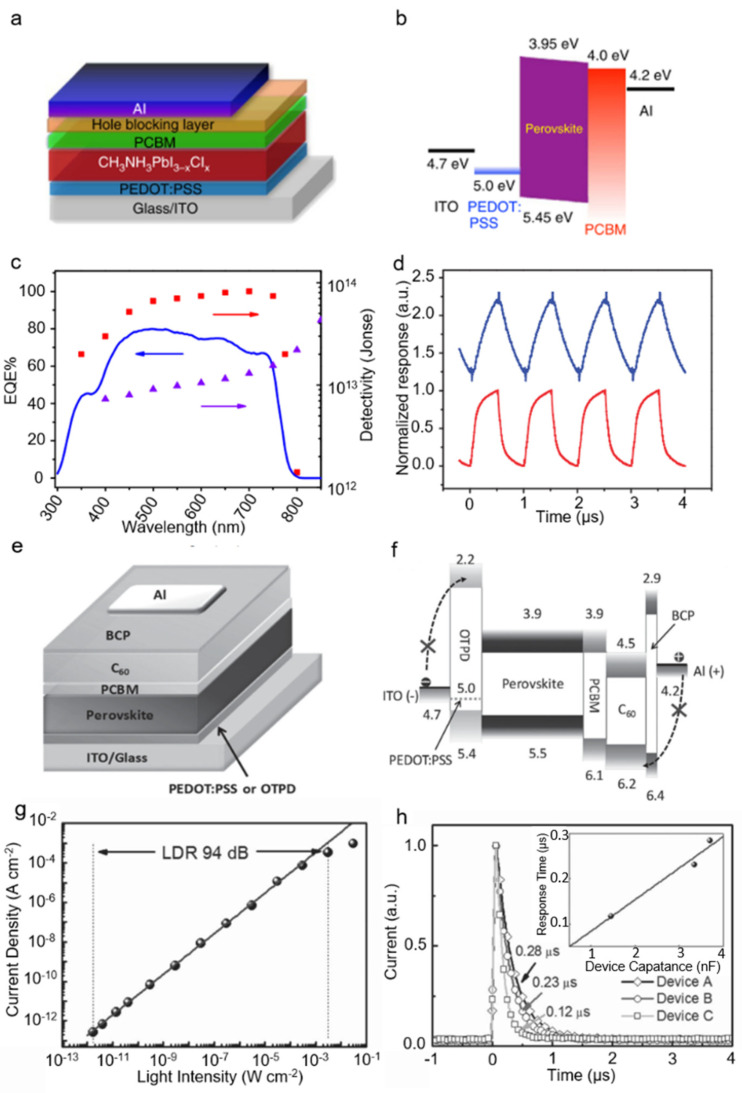
(**a**) The schematic of the photovoltaic device. (**b**) Energy diagram of the device. (**c**) EQE and detectivity of the photodetector at different wavelength. (**d**) Transient photocurrent response of the photodetector at a pulse frequency of 1 MHz with a device area of 0.1 cm^2^ (blue line) and 0.01 cm^2^ (red line), respectively. Copyright 2014, Nature Publishing Group [80] (**e**) The schematic of perovskite photodetectors. (**f**) The energy diagram of the perovskite photodetectors. (**g**) Dynamic response of the photodetector. (**h**) The short photoresponse time of the device. Reprinted with permission from ref. [81]. Copyright 2015 Wiley-VCH.

**Figure 33 nanomaterials-11-01038-f033:**
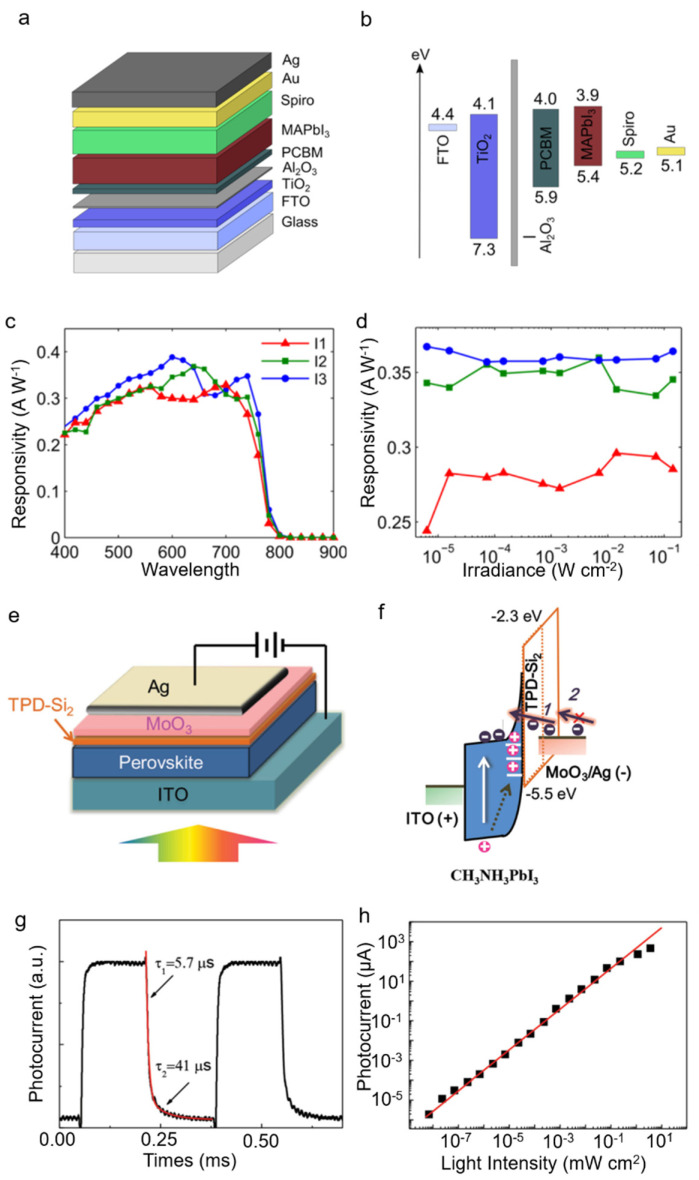
(**a**) The schematic of the photovoltaic device. (**b**) Energy diagram of the device. (**c**) Broadband response spectra of the device. (**d**) Responsivity versus incident light with different intensities. Copyright 2015, American Chemical Society [83]. (**e**) The schematic of the MoO_3_-MAPbI_3_ photodetector. (**f**) Energy diagram of the photodetector. (**g**) Transient photocurrent of the MAPbI_3_ photodetector. (**h**) Dynamic response of the photodetector. Reprinted with permission from ref. [86]. Copyright 2015 Wiley-VCH.

**Table 1 nanomaterials-11-01038-t001:** The key parameters of photodetectors.

Parameters	Definition
Photoresponsivity (R)	the ratio of the photocurrent to the incident power on the active area: R = (I_p_ − I_d_)/(PA), where I_p_ is the photocurrent, I_d_ is the dark current, P is the light intensity, A is the active area
EQE	Photoelectric conversion efficiency. EQE = Rhc/e, where h is the Planck’s constant, c is the light velocity, e is the electronic charge.
Gain (G)	The number of charge carriers through external circuit for per incident photon: G = τ_l_/τ_t_ = τ_l_(µV)/d^2^, where τ_l_ is the carrier lifetime, τ_t_ is the carrier transit time, µ is the carrier mobility, V is bias voltage, d is the channel length
Detectivity (D*)	D* = (A△ f)^1/2^/NEP, where A is the active area of the detector, △f is the electrical bandwidth, NEP is the noise equivalent power.
LDR	LDR usually stands for “Linear Dynamic Range”, defined as the range in which the current response of the photodetector is linearly proportional to the light intensity. LDR = 20 log (I_p_*/I_d_), Where I_d_ is the dark current.
Response speed (rise/decay time)	The ability of devices to track the incident light signal.

**Table 2 nanomaterials-11-01038-t002:** Key characteristics of nanostructured perovskites-based photodetectors.

Device	Dimension	Perovskite	R	D	EQE	On/Off Ratio	Gain	LDR	Rise/Decay Time	Ref
Photoconductor	2D	CsPbBr_3_	0.25		53%				19 µs/25 µs	[43]
		MAPbI_3_				1210				[53]
		CsPbBr_3_/CNTs	31.1		7488%				16 µs/0.38 ms	[55]
	1D	MAPbI_3_	13.57	5.25 × 10^12^						[56]
		MAPbI_3_	0.04	0.6 × 10^12^		0.84 × 10^4^			178 µs/173 µs	[57]
		MAPbBr_3_				61.9			0.12 s/0.086 s	[58]
		MAPb(I_1-x_Br_x_)_3_	1.25 × 10^4^	1.73 × 10^11^			36,800	150		[59]
		MAPb(I/Br)_3_	10^3^			2000			0.68 s/0.66 s	[60]
		MAPbI_3_	0.45	2 × 10^13^		4000			<0.1 ms	[61]
		MAPbI_3_	1.2	2.39 × 10^12^		160				[62]
		CsPbI_3_	1294	2.6 × 10^14^					0.85 ms/0.78 ms	[42]
		CsSnI_3_	0.054	3.85 × 10^5^					83.8 ms/243.4 ms	[63]
		CsCu_2_I_3_	32.3	1.89 × 10^12^		2.6 × 10^3^			6.94 µs/214 µs	[64]
	others	CsPbBr_3_	0.0209			1.6 × 10^5^			0.2 ms/1.3 s	[65]
		CsPbX_3_				10^5^			24 ms/29 ms	[52]
		GQDs/CsPbBr_3_	0.24	2.5 × 10^12^	57%	7.2 × 10^4^			1.16 ms (Decay)	[66]
Phototransistor	2D	MAPbX_3_	22			10^12^			20 ms/40 ms	[67]
		CsPbBr_3_							17.8 ms/(14.7 ms/15.2 ms)	[68]
	1D	MAPbI_3_							0.35 ms/0.25 ms	[69]
		MAPbI_3_	0.3			4.02 × 10^3^			0.2 ms/0.25 ms	[70]
		MAPbI_3_	0.11				0.25			[71]
		MAPbI_3_	2.6 × 10^6^						55 s/75 s	[72]
		C8BTBT/CsPbI_3_	4.3 × 10^3^			2.2 × 106^6^				[73]
		CsPbX_3_	4489	7.9 × 10^12^					<50 ms	[74]
		CsPbI_3_	2.92 × 10^3^	5.17 × 10^13^	0.6 × 10^6^%				50ms	[75]
		CsPbI_3_	1.84 × 10^3^	9.9 × 10^13^	5.65 × 10^5^%				3.78 ms/539 ms	[76]
	others	Graphene/CsPbBr_3-x_I_x_	8.2 × 10^8^	2.4 × 10^16^					0.81 s/3.65 s	[77]
		Graphene/CsPbBr_x_I3-_x_	1.12 × 10^5^	1.17 × 10^11^			9.32 × 10^10^		273.6 ms/2.26 s	[78]
		CsPbI_3-x_Br_x_	7.7 × 10^4^	5.6 × 10^11^	107%				0.59 s/0.32 s	[44]
		CsPbI_3_/DPP-DTT	110	2.9 × 10^13^		6 × 10^3^				[79]

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
