# Peer review of "High-Performance Photodetectors Based on Nanostructured Perovskites"

_nanomaterials, 2021, doi:10.3390/nano11041038_

Round 1

Reviewer 1 Report

The submitted manuscript presents a potentially interesting review on photodetectors based on nanostructured perovskites. This review fits well with the aims of the journal but requires some revisions before it can be considered acceptable for publication.

(1) Introduction: the first paragraph provides some instructions for authors and should be deleted.

(2) Lines 57-58: “Recently, perovskites with a typical formula of ABX3 has attracted wide research interest in the photodetector field.” Appropriate references are necessary.

(3) I would suggest adjusting the layout of table 1 slightly. Please separate the rows to make it easier to understand the information in the second column.

(4) In my opinion, the “Introduction” section needs to be reorganized to focus on nanostructured perovskites, as they are the objects of research.

(5) Insert a link to figure 1 in the text.

(6) The style used by the authors is too descriptive. Obviously, there are now many published works on the subject but I would recommend authors to add a bit more critical insights.

(7) It would be very important if the authors could provide, possibly in the section “Conclusions”, more critical opinions about the possible future evolution of the research field.

Author Response

Response to Reviewer 1 Comments

Point 1: Introduction: the first paragraph provides some instructions for authors and should be deleted. 

Response 1: Thank you for your advice. The first paragraph has been deleted.

Point 2: Lines 57-58: “Recently, perovskites with a typical formula of ABX3 has attracted wide research interest in the photodetector field.” Appropriate references are necessary.

Response 2: Thank you for your suggestion. We added 4 references as follow:

Recently, perovskites with a typical formula of ABX3 has attracted wide research interest in the photodetector field. [23-26]

  1. Fang, Y.; Huang, J. Resolving weak light of sub‐picowatt per square centimeter by hybrid perovskite photodetectors enabled by noise reduction. Advanced materials 2015, 27, 2804-2810.
  2. Bao, C.; Zhu, W.; Yang, J.; Li, F.; Gu, S.; Wang, Y.; Yu, T.; Zhu, J.; Zhou, Y.; Zou, Z. Highly flexible self-powered organolead trihalide perovskite photodetectors with gold nanowire networks as transparent electrodes. ACS applied materials & interfaces 2016, 8, 23868-23875.
  3. Lin, Q.; Armin, A.; Burn, P.L.; Meredith, P. Filterless narrowband visible photodetectors. Nature Photonics 2015, 9, 687-694.
  4. Maculan, G.; Sheikh, A.D.; Abdelhady, A.L.; Saidaminov, M.I.; Haque, M.A.; Murali, B.; Alarousu, E.; Mohammed, O.F.; Wu, T.; Bakr, O.M. CH3NH3PbCl3 single crystals: inverse temperature crystallization and visible-blind UV-photodetector. The journal of physical chemistry letters 2015, 6, 3781-3786.

Point 3: I would suggest adjusting the layout of table 1 slightly. Please separate the rows to make it easier to understand the information in the second column. 

Response 3: Thank you for your advice. The table was adjusted as follow.

Parameters

Definition

Photoresponsivity (R)

the ratio of the photocurrent to the incident power on the active area: R = (Ip-Id)/(PA), where Ip is the photocurrent, Id is the dark current, P is the light intensity, A is the active area

EQE

Photoelectric conversion efficiency. EQE = Rhc/e, where h is the Planck’s constant, c is the light velocity, e is the electronic charge.

Gain (G)

The number of charge carriers through external circuit for per incident photon: G = τlt = τl(µV)/d2, where τl is the carrier lifetime, τt is the carrier transit time, µ is the carrier mobility, V is bias voltage, d is the channel length

Detectivity (D*)

D* = (A△f)1/2/NEP, where A is the active area of the detector, △f is the electrical bandwidth, NEP is the noise equivalent power.

LDR

LDR usually stands for “Linear Dynamic Range”, defined as the range in which the current response of the photodetector is linearly proportional to the light intensity. LDR = 20 log (Ip*/Id),Where Id is the dark current.

Response speed (rise/decay time)

The ability of devices to track the incident light signal.

Point 4:  In my opinion, the “Introduction” section needs to be reorganized to focus on nanostructured perovskites, as they are the objects of research. 

Response 4: Thank you for your advice. To further highlight the focus of this article, We added the following.

Line 79: For example, a high responsivity of 1294 A W−1 with a ultrahigh detectivity of 2.6 × 1014 Jones was obtained in α-CsPbI3 nanowire-based photodetector. [45] In addition, an ultrahigh response speed (19/25 μs) was obtained in a photodetector based on atom-thin 2D CsPbBr3 nanosheets. [46] An ultrahigh EQE over 107% was demonstrated by a phototransistor based on CsPbI3-xBrx quantum dots (QDs)/monolayer MoS2 heterostructure. [47] All these are enough to show that the photodetectors based on nanostructured perovskites have more advantages in ultrahigh responsivity and ultrafast response speed. There have been some reviews on nanostructured perovskite-based photodetectors. Gu et al.[48] focus on the effect of elemental composition and dimensionality of the perovskite materials on photodetector performance. Wang et al.[49] systematically summarized the synthesis, optoelectronic properties, and performance of photodetectors based on low-dimensional perovskites. Here, more emphasis is placed on the effect of the device structure and the dimension of nanostructured perovskites on the device performance.

Point 5: Insert a link to figure 1 in the text.

Response 5: Thank you for your advice. We insert links of figure 1 in the corresponding position.

Line 109: For photovoltaic photodetector, or photodiode (Figure 1a), the device structure is similar to that of solar cell configuration.

Line 116: As for photoconductive photodetector, it can be further divided into photoconductor (Figure 1b) and phototransistor (Figure 1c).

Point 6: The style used by the authors is too descriptive. Obviously, there are now many published works on the subject but I would recommend authors to add a bit more critical insights. 

Response 6: Thank you for your advice. We added some critical content to make the article less descriptive.

For example:

Line 181: The high response speed indicates the rapid separation and efficient extraction of photogenerated carriers, which can be attributed to the improved electrical conductivity of CNTs.

Line 196: The outstanding device performance can be attributed to the high optical absorption coefficient of MAPbI3 and high crystallinity of the MWs.

Line 238: The high performace could be attributed to the long carrier lifetime and high carrier mobility in high-crystalline MAPbI3 NWs. Thus, a variety of high-performance integrated optoelectronic devices could be fabricated by the NW arrays.

Line 267: The improved performance can be attributed to the increasing of carrier lifetime after OA passivation, which can reduces the non-radiating composite centers on the surface of the NWs and gives the device more time to collect and transfer the photogenerated carriers.

Point 7: It would be very important if the authors could provide, possibly in the section “Conclusions”, more critical opinions about the possible future evolution of the research field. 

Response 7: Thank you for your advice. Some critical contents were added in the “Conclusions”.

Line 647: In addition, some literatures only reported the best performance of the device, but ignored the average performance of the device, which would lead to a misleading effect on the industrialization direction, and also show that the authors have no confidence in the stability of the device.

Reviewer 2 Report

There are some issues that need to be solved.

  1. The abstract is not clear. Please provide some concrete achievements of this study.
  2. There are two periods at the end of the abstract.
  3.  The first graph of the introduction is very strange. Please check it.
  4. English needs to be improved. Grammatical errors need to be fixed.
  5. The authors present various types of PDs in this review paper. However, physics and mechanisms for the improvements  or advantages are lack. 
  6. The authors should make a table to juxtapose all the different types of PDs with different parameters. In addition, figure of merit should be provided to evaluate different devices.

7 .The authors should provide more perovskite-based PD references within 3 years.  

Author Response

Point 1: The abstract is not clear. Please provide some concrete achievements of this study.

Point 2: There are two periods at the end of the abstract. 

Response: Thank you for your advice. Since the first and second questions are about abstract, we have rewritten the abstract as follow:

In recent years, high-performance photodetectors have attracted wide attentions because of their important applications including imaging, spectroscopy, fibre-optic communications, remote control, chemical/biological sensing and so on. Nanostructured perovskites are extremely suitable for detective applications with their long carrier lifetime, high carrier mobility, facile synthesis and beneficial to device miniaturization. Because the structure of the device and the dimension of nanostructured perovskite have a profound impact on the performance of photodetector, we divide nanostructured perovskite into 2D, 1D and 0D, and review their applications in photodetector (including photoconductor, phototransistor and photodiode) respectively. The devices exhibit high performance with high photoresponsivity, large EQE, large gain, high detectivity and fast response time. The intriguing properties suggest that nanostructured perovskites have a great potential in photodetection.

Point 3: The first graph of the introduction is very strange. Please check it.

Response: Thank you for your advice. The first paragraph provides some instructions for us and we deleted it in the revised article.

Point 4: English needs to be improved. Grammatical errors need to be fixed.

Response: Thank you for your advice. We corrected the grammatical errors in the revised article.

Point 5: The authors present various types of PDs in this review paper. However, physics and mechanisms for the improvements or advantages are lack.

Response: Thank you for your advice. Some mechanisms for advantages have been mentioned in the original manuscript, for example:

Line 105: “vertical photodetectors provide fast response and low driving voltage because of the small electrode spacing with a short carrier transit length; in contrast, lateral photodetectors show slow response and high driving voltage due to their large electrode spacing”

Line 119: “External voltage leading to multiple electrical carriers recycling per single incident photon should be responsible for the large gain”

Line 120: “large gain, in turn, usually results in a slow response speed because both the response time and the gain are determined by the carrier lifetime”

 In addition to the existing mechanism explanation, we add a new explanation about the detection performance of photodiode as follow:

Line 114: Owing to the junction barrier at the interface, photodiodes exhibit low dark current and large detectivity.  

Point 6: The authors should make a table to juxtapose all the different types of PDs with different parameters. In addition, figure of merit should be provided to evaluate different devices.

Response: Thank you for your advice. We make the table to exhibit different types of PDs with different parameters.

Table 2. Key characteristics of nanostructured perovskites-based photodetectors

Device

Dimension

Perovskite

R

D

EQE

On/Off ratio

Gain

LDR

rise/decay time

Ref

Photoconductor

2D

CsPbBr3

0.25

53%

19μs /25μs

46

MAPbI3

1210

56

CsPbBr3/CNTs

31.1

7488%

16μs/0.38ms

58

1D

MAPbI3

13.57

5.25×1012

59

MAPbI3

0.04

0.6×1012

0.84×104

178μs /173μs

60

MAPbBr3

61.9

0.12s/0.086s

61

MAPb(I1-xBrx)3

1.25×104

1.73×1011

36800

150

62

MAPb(I/Br)3

103

2000

0.68s/0.66s

63

MAPbI3

0.45

2×1013

4000

<0.1ms

64

MAPbI3

1.2

2.39×1012

160

65

CsPbI3

1294

2.6×1014

0.85ms/0.78ms

45

CsSnI­3

0.054

3.85×105

83.8ms/243.4ms

66

CsCu2I3

32.3

1.89×1012

2.6×103

6.94μs/214μs

67

others

CsPbBr3

0.0209

1.6×105

0.2ms/1.3s

68

CsPbX3

105

24ms/29ms

55

GQDs/CsPbBr3

0.24

2.5×1012

57%

7.2×104

1.16ms(Decay)

69

Phototransistor

2D

MAPbX3

22

1012

20ms/40ms

70

CsPbBr3

17.8ms/(14.7ms/15.2ms)

71

1D

MAPbI3

0.35ms/0.25ms

72

MAPbI3

0.3

0.2ms/0.25ms

73

MAPbI3

0.11

0.25

74

MAPbI3

2.6×106

55s/75s

75

C8BTBT/CsPbI3

4.3×103

2.2×106

76

CsPbX3

4489

7.9×1012

<50ms

77

CsPbI3

2.92×103

5.17×1012

0.6×106%

78

CsPbI3

1.84×103

9.9×1013

5.65×105%

3.78ms/539ms

79

others

Graphene/CsPbBr3-xIx

8.2×108

2.4×1016

0.81s/3.65s

80

Graphene/CsPbBrxI3-x

1.12×105

1.17×1011

9.32×1010

273.6ms/2.26s

81

CsPbI3-xBrx

7.7×104

5.6×1011

107%

0.59s/0.32s

47

CsPbI3/DPP-DTT

110

2.9×1013

6×103

82

Point 7: The authors should provide more perovskite-based PD references within 3 years. 

Response: Thank you for your advice. We added the perovskite-based PD references within 3 years as follow:

Line 482: Meng et al. fabricated a phototransistor based on CsPbX3 (X = Cl, Br, or I) NWs with a uniform diameter of ∼150 nm. These devices exhibit high performance with the responsivity exceeding 4489 A/W and detectivity over 7.9 × 1012 Jones. The response times are found to be less than 50 ms. The excellent performance can be attributed to the reduced defect concentration in CsPbX3 NWs as well as the field-effect transistors (FET) with superior hole field-effect mobility of 3.05 cm2/(V s).[77]

Figure 25. (a-b) Responsivity and detectivity vs light intensity. (c-d) I-t curve under illumination (9 mW/cm2) and a bias voltage of 5V. Copyright 2019, American Chemical Society.[77]

Yang et al. constructed a phototransistor based on CsPbI3 nanorods. The schematic diagram of the device was shown in Figure 26a. The linear and symmetric I-V curves demonstrated that the contact was ohmic, as shown in Figure 26(b). The as-fabricated device exhibited a totally excellent performance, such as high responsivity of 2.92 × 103 A·W-1, large EQE of 0.9 × 106 %, fast response time of 0.05ms and a high detectivity of 5.17 × 1013Jones (Figure 26c-f). The excellent performance is mainly due to the following two reasons. First, high absorption coefficient, low recombination of charge carriers and low density of defects of CsPbI3 nanorods generate strong photoelectric effect. Second, the high-quality nanorod provides a smooth and short path for carrier transfer, and significantly improves the response speed. [78]

Figure 26. (a) Schematic diagram of the as-fabricated phototransistor based on a single CsPbI3 nanorod. (b) I−V characteristics under dark and 405 nm irradiation. (c-d) Representative, EQE and Detectivity vs wavelength. (e-f) I-t curve and response time of the photodetector. Copyright 2018, American Chemical Society. [78]

Du et al. developed a phototransistor based on CsPbI3 nanotubes, which can be stable for more than 2 months under air conditions. The schematic diagram was shown in Figure 27a. The phototransistor exhibited an excellent performance with an EQE, detectivity, photoresponsivity and response time of 5.65 × 105 %, 9.99 × 1013 Jones, 1.84 × 103 A W−1 and 3.78 ms/359 ms, respectively (Figure 27 b-d). It is comparable to the best of all inorganic perovskite photodetectors, which is mainly attributed to the enhanced light absorption resulting from the light trapping effect within the tube cavity.[79]

Figure 27. The schematic diagram for the phototransistor based on CsPbI3 nanotubes. (b) Responsivity and EQE vs incident intensity. (c) Responsivity and Detectivity vs drain voltage. (d) Response time of the device. Copyright 2019, Wiley-VCH. [79]

Reviewer 3 Report

The manuscript focuses on a very interesting topic, due to the appealing properties of nanostructured perovskite-based materials for optoelectronic devices, in particular for photodetectors. The relevance and originality of a review of this topic should therefore be specially highlighted in the present context.

The authors state that photodiodes based on nanostructured perovskites are rarely reported. However, there are some recent review articles that should be referenced and discussed in the appropriate section. For example: “Low‐Dimensional Metal Halide Perovskite Photodetectors”, by Wang et al., published in Advanced Materials (https://doi.org/10.1002/adma.202003309); or “Emerging Perovskite Materials with Different Nanostructures for Photodetectors”, published last year in Advanced Optical Materials.  In fact, the latter is included in this submission as reference [41] but, in my opinion, it should be commented as the comprehensive review it is.

In order to highlight the novel contributions of the present manuscript, it would be necessary to comment on the central points of recent review articles (and, where appropriate, the aspects not included in them)

In addition to the previous concern, a number of formal aspects (not only misprints) must be revised and corrected. I list a number of them (by way of example) but the manuscript needs to be reviewed in depth:

  1. At the beginning of the Introduction section, the “Guidelines for authors” must be removed.
  2. The definitions in Table 1 must be revised; some of them are not precisely expressed or seem to be incomplete: i.e. the grammar structure of the definition of Responsivity must be corrected and Ip must be defined; LDR usually stands for “Linear Dynamic Range”, defined as the range in which the current response of the photodetector is linearly proportional to the light intensity; Id must be defined…
  3. The numbering of references in the text (and in the captions for figures) do not correspond with the reference list (at least from ref. [47] onwards, the reference numbers in the text must be increased in one unit).
  4. There are inconsistencies in some figures (axis labelling, some captions, etc.). Some of them come from the original papers but, in my opinion, they should be corrected, or at least noted, in a review article. Some examples:
    1. In figure 9 e), the X axis label is not correct: the current is represented vs voltage (V), no vs Time (S).
    2. In figure 12, the title of graph b) is not correct; the graph does not show photoluminescence spectra but the intensity decay over time.
    3. In figure 21, the caption for graph b) is missing.
    4. The label of X axis in figure 24 b) is not visible.
  5. There are other grammar or spelling issues to be corrected. I list a few as an example, but the manuscript needs to be revised.
    1. Some chemical formulas must be corrected (subscripts). For example, in page 6, in the caption for figure 3: MAPbI3; in page 7, in the caption for figure 5 and in line 204: MAPbI3; in page 12, in the caption for figure 12: CsPbI3;…
    2. The superscript label “3”,  in the authors´affiliation list, is not actually assigned to any of the co-authors.
    3. The meaning of some acronyms should be indicated; for example, OA, in page 10, line 256.
    4. In page 7, line 204: In the sentence “It is worth noting that the performance of the as-fabricated device has a significantly improved” must be corrected; In page 11, line 288: In the sentence “The rise time is 0.85 ms, and the decay time is 0.78  ms, respectively”, the word respectively should be removed;  In page 13, line 319: the concordance in the sentence “An asymmetrical I-V curves demonstrated…” must be corrected; In page 18, line 411:  the sentence “The linear output characteristics indicate that the ohmic contacts ...” must be reformulated…

Author Response

The authors state that photodiodes based on nanostructured perovskites are rarely reported. However, there are some recent review articles that should be referenced and discussed in the appropriate section. For example: “Low‐Dimensional Metal Halide Perovskite Photodetectors”, by Wang et al., published in Advanced Materials (https://doi.org/10.1002/adma.202003309); or “Emerging Perovskite Materials with Different Nanostructures for Photodetectors”, published last year in Advanced Optical Materials.  In fact, the latter is included in this submission as reference [41] but, in my opinion, it should be commented as the comprehensive review it is.

In order to highlight the novel contributions of the present manuscript, it would be necessary to comment on the central points of recent review articles (and, where appropriate, the aspects not included in them)

Response: Thank you for your advice. The article was revised as follow:

Line 86: There have been some reviews on nanostructured perovskite-based photodetectors. Gu et al.[48] focus on the effect of elemental composition and dimensionality of the perovskite materials on photodetector performance. Wang et al.[49] systematically summarized the synthesis, optoelectronic properties, and performance of photodetectors based on low-dimensional perovskites. Here, more emphasis is placed on the effect of the device structure and the dimension of nanostructured perovskites on the device performance.

Point 1: At the beginning of the Introduction section, the “Guidelines for authors” must be removed. 

Response 1: Thank you for your advice. The “Guidelines for authors” has been deleted.

Point 2: The definitions in Table 1 must be revised; some of them are not precisely expressed or seem to be incomplete: i.e. the grammar structure of the definition of Responsivity must be corrected and Ip must be defined; LDR usually stands for “Linear Dynamic Range”, defined as the range in which the current response of the photodetector is linearly proportional to the light intensity; Id must be defined…

Response 2: Thank you for your advice. We revised the table 1 as follow:

Parameters

Definition

Photoresponsivity (R)

the ratio of the photocurrent to the incident power on the active area: R = (Ip-Id)/(PA), where Ip is the photocurrent, Id is the dark current, P is the light intensity, A is the active area

EQE

Photoelectric conversion efficiency. EQE = Rhc/e, where h is the Planck’s constant, c is the light velocity, e is the electronic charge.

Gain (G)

The number of charge carriers through external circuit for per incident photon: G = τlt = τl(µV)/d2, where τl is the carrier lifetime, τt is the carrier transit time, µ is the carrier mobility, V is bias voltage, d is the channel length

Detectivity (D*)

D* = (A△f)1/2/NEP, where A is the active area of the detector, △f is the electrical bandwidth, NEP is the noise equivalent power.

LDR

LDR usually stands for “Linear Dynamic Range”, defined as the range in which the current response of the photodetector is linearly proportional to the light intensity. LDR = 20 log (Ip*/Id),Where Id is the dark current.

Response speed (rise/decay time)

The ability of devices to track the incident light signal.

Point 3: The numbering of references in the text (and in the captions for figures) do not correspond with the reference list (at least from ref. [47] onwards, the reference numbers in the text must be increased in one unit).

Response 3: Thank you for helping us find this problem. We have modified the serial number of the document.

Point 4: There are inconsistencies in some figures (axis labelling, some captions, etc.). Some of them come from the original papers but, in my opinion, they should be corrected, or at least noted, in a review article. Some examples:

  1. In figure 9 e), the X axis label is not correct: the current is represented vs voltage (V), no vs Time (S).
  2. In figure 12, the title of graph b) is not correct; the graph does not show photoluminescence spectra but the intensity decay over time.
  3. In figure 21, the caption for graph b) is missing.
  4. The label of X axis in figure 24 b) is not visible.

Response 4: Thank you very much for your careful review. We revised the article as follow:

1) Figure 9 was revised as follow:

2) We don't think the illustration of Figure 12b as “time-resolved PL spectra” is wrong.

3) We added the illustration of  Figure 21b as follow:

(b) I-V curve of the photodetector under dark and illumination at 100 mWcm-2.

4) Figure 24 was revised as follow:

Point 5:  There are other grammar or spelling issues to be corrected. I list a few as an example, but the manuscript needs to be revised.

  1. Some chemical formulas must be corrected (subscripts). For example, in page 6, in the caption for figure 3: MAPbI3; in page 7, in the caption for figure 5 and in line 204: MAPbI3; in page 12, in the caption for figure 12: CsPbI3;…
  2. The superscript label “3”,  in the authors´affiliation list, is not actually assigned to any of the co-authors.
  3. The meaning of some acronyms should be indicated; for example, OA, in page 10, line 256.
  4. In page 7, line 204: In the sentence “It is worth noting that the performance of the as-fabricated device has a significantly improved” must be corrected; In page 11, line 288: In the sentence “The rise time is 0.85 ms, and the decay time is 0.78  ms, respectively”, the word respectively should be removed;  In page 13, line 319: the concordance in the sentence “An asymmetrical I-V curves demonstrated…” must be corrected; In page 18, line 411:  the sentence “The linear output characteristics indicate that the ohmic contacts ...” must be reformulated…

Response 5: Thank you for your advice. We have corrected these errors and further revised other grammatical errors in the article.

  1. We rechecked and revised the subscripts in the article.
  2. The superscript label “3”,  in the authors´affiliation list, should be assigned to the second author Jie Li.
  3. We revised the acronyms as follow:

OA (Oleic acid)-passivated

  1. We revised these mistakes as follow:

line 204: It is worth noting that the performance of the as-fabricated device has been significantly improved

line 288: The rise time is 0.85 ms, and the decay time is 0.78  ms.

line 319: The asymmetrical I-V curves demonstrated the formation of a Schottky barrier

line 411: The linear output characteristics indicate that the contacts between MAPbI3 NWs and electrodes are ohmic.

Round 2

Reviewer 1 Report

I think, that the manuscript can be published in the journal.

Reviewer 3 Report

I consider the revised version adequate for publication.